# Spatio-Angular Convolutions for Super-resolution in Diffusion MRI

**Matthew Lyon**
University of Manchester
matthew.s.lyon@.manchester.ac.uk

**Paul Armitage**
University of Sheffield
p.armitage@sheffield.ac.uk

**Mauricio A Álvarez**
University of Manchester
mauricio.alvarezlopez@manchester.ac.uk

## Abstract

Diffusion MRI (dMRI) is a widely used imaging modality, but requires long scanning times to acquire high resolution datasets. By leveraging the unique geometry present within this domain, we present a novel approach to dMRI angular super-resolution that extends upon the parametric continuous convolution (PCConv) framework. We introduce several additions to the operation including a Fourier feature mapping, global coordinates, and domain specific context. Using this framework, we build a fully parametric continuous convolution network (PCCNN) and compare against existing models. We demonstrate the PCCNN performs competitively while using significantly fewer parameters. Moreover, we show that this formulation generalises well to clinically relevant downstream analyses such as fixel-based analysis, and neurite orientation dispersion and density imaging.

## 1 Introduction

Diffusion magnetic resonance imaging (dMRI) is a clinical imaging modality that is routinely used to diagnose white matter pathology within the human brain. Advancements in downstream analyses using dMRI data continuously expand the diagnostic capabilities of the modality, but often require high resolution data that is clinically infeasible to acquire [28]. However, recent research demonstrates that deep learning can be used to predict high resolution images using low resolution dMRI data [26, 44, 41], a technique known as super-resolution. These advancements therefore pave the way for clinically accessible advanced dMRI analyses.

dMRI data can be represented using a non-Euclidean geometry. Specifically, a dMRI dataset consists of a series of three-dimensional volumes where each volume measures the water diffusivity in a given direction. Here, the direction is quantified by a 3D vector known as the 'b-vector', and is sparsely sampled from a unit sphere. Each three-dimensional volume, however, is regularly sampled in a discrete grid. Therefore, dMRI data has two distinct resolution types: *spatial* resolution which denotes the regularly sampled grid size, and *angular* resolution which denotes the number of diffusion directions. Despite this relatively unique geometry, most dMRI deep learning methods use operations grounded in a Euclidean framework, such as convolutional neural networks (CNNs) [43, 44, 41].

This presents an opportunity as typical CNN architectures do not fully utilise the geometric properties present in dMRI data. For example, implicit within the formulation of the CNN is the assumption that data are densely and regularly sampled in a discrete manner. This is only true when considering the three spatial dimensions of dMRI data, whilst the other dimensions would be more suited using approaches like graph convolutional networks (GCNs) [46], spherical CNNs [7], and point cloud CNNs [22]. Examples of approaches that develop geometrically motivated convolutions in the dMRI

37th Conference on Neural Information Processing Systems (NeurIPS 2023).

domain include those for tissue segmentation [21], lesion segmentation [29], and fibre orientation distribution (FOD) reconstruction [5, 32].

Another promising candidate for a framework capable of utilising the unique dMRI geometry is the parametric continuous convolution (PCConv) introduced by Wang et al. [39], which extended the discrete convolution into the continuous domain. This was achieved via an inner parameterised hypernetwork [16] responsible for generating convolutional weights. Here, the convolutional weights were not learnt directly but were instead sampled from the hypernetwork given kernel coordinates as input. This operation could therefore be used to model arbitrary geometries.

In this work, we build upon the PCConv operation to convolve across both spatial and angular dimensions of dMRI data via a new, flexible, coordinate embedding. We subsequently create a deep parametric continuous CNN (PCCNN) that is well-suited for inference within this domain. Furthermore, we develop additional modifications to the PCCNN by including supplemental prior information in the coordinate embedding. These modifications include the application of Fourier feature mappings, the inclusion of prior domain information, and the incorporation of global coordinate components.

We demonstrate the effectiveness and flexibility of this approach through the task of angular super-resolution of dMRI data in various sampling schemes, including single-shell and multi-shell data [18]. To assess the performance of our method, we compare results across clinically relevant downstream analysis tasks such as fixel-based analysis (FBA) [30], and neurite orientation dispersion and density imaging (NODDI) [45]. We show that the PCCNN performs competitively against a similar recurrent CNN (RCNN) network trained on the same data whilst using approximately a tenth of the number of parameters. Additionally, we demonstrate the generalisability of the network by comparing it against models trained specifically for downstream analysis tasks.

## 2    Related Works

Whilst there is a body of literature within *spatial* super-resolution methods in the natural image domain [40], and more specifically the dMRI domain [1, 36, 6, 25], this work focuses on *angular* super-resolution as a means of increasing image quality. Here, the task of inferring high angular resolution data from low angular resolution dMRI acquisitions can broadly be split into two categories: methods that super-resolve raw dMRI data, and those that super-resolve downstream analyses. The former allows for more flexible use of the data and analysis method, but the unconstrained nature of the task makes inference more challenging. The latter constrains the problem by leveraging assumptions made within the analysis method, however the mapping from raw dMRI data to downstream analysis is in general not invertible, therefore the super-resolved data cannot be used to derive other downstream analysis metrics.

Notable works in angular super-resolution of raw dMRI data include Yin et al. [43], whereby a preliminary study constructed a 1D CNN autoencoder that only convolved across the angular dimension, disregarding possible spatial correlations within the data. Lyon et al. [26] developed a 3D RCNN that benefited from parameter sharing in the spatial dimensions, but required angular dimension shuffling to overcome sequence order bias. Both models concatenated the b-vectors as channels within the input, therefore treating the b-vectors as additional features, rather than coordinates of a manifold for which the data lies on. Finally, Ren et al. [31] proposed a conditional generative adversarial network (GAN) that additionally used structural T1w and T2w data, as well as b-vector information to predict unseen dMRI intensity data.

As there exists a myriad of downstream analyses for dMRI, this work focuses on two commonly used methods: FBA [30] and NODDI [45]. FBA relies on segmenting FODs [18] into 'fixels' or fibres within a voxel. Relevant works in FOD super-resolution include Lucena et al. [24] and Zeng et al. [44]. Both of these studies regressed low angular resolution derived FODs with their high resolution counterparts through 3D CNNs. NODDI estimates micro-structural complexity of the underlying brain tissue through modelling axon and dendrite populations within a voxel. Approaches to infer high angular resolution derived NODDI parameters from lower resolution raw dMRI data include Golkov et al. [15], Ye et al. [41], and Ye et al. [42]. These studies use various sub-sampling schemes and network architectures, including a 3D CNN within Ye et al. [41].

# 3 Methods

This section introduces parametric continuous convolutions, and extends the framework into the dMRI domain. We highlight our key contributions to the PCConv operation, then finally provide implementation details used to perform training and inference within the task of dMRI super-resolution.

## 3.1 Parametric Continuous Convolutions

A discrete convolution within the context of deep learning is defined as follows

$$h[n] = (f * g)[n] = \sum_{m=-M}^{M} f[m]g[n-m],$$

where the data function $f : \mathcal{N} \to \mathbb{R}$ and kernel $g : \mathcal{G} \to \mathbb{R}$ are defined over the support domain for the finite integer set $\mathcal{N} = \mathbb{Z}^D$ and $\mathcal{G} = \{-M, -M+1, \dots, M-1, M\}^D$ respectively. Conversely, a continuous convolution is defined as

$$h(\mathbf{y}_j) = (f * g)(\mathbf{y}_j) = \int_{-\infty}^{\infty} f(\mathbf{x})g(\mathbf{y}_j - \mathbf{x})d\mathbf{x},$$

where both $f$ and $g$ are continuous functions over the support domain $\mathcal{N} = \mathcal{G} = \mathbb{R}^D$. Typically in deep learning, this integration is analytically intractable but can be approximated as in Wang et al. [39] using the following form:

$$h(\mathbf{y}_j) = \int_{-\infty}^{\infty} f(\mathbf{x})g(\mathbf{y}_j - \mathbf{x})d\mathbf{x} \approx \frac{1}{N} \sum_{i=1}^{N} f(\mathbf{x}_i)g(\mathbf{y}_j - \mathbf{x}_i), \tag{1}$$

where a finite number of input points $\mathbf{x}_i$ are sampled from the support domain for each output point $\mathbf{y}_j$. In the parametric continuous convolution framework, we choose to use a multi-layer perceptron (MLP) as the kernel function $g(\mathbf{e}; \boldsymbol{\theta}) = \text{MLP}(\mathbf{e}; \boldsymbol{\theta})$, where $\mathbf{e}$ is a coordinate embedding that depends on $\mathbf{x}_i$ and $\mathbf{y}_j$, which is described in detail in Section 3.2. The kernel function $g(\mathbf{e}; \boldsymbol{\theta}) : \mathbb{R}^D \to \mathbb{R}$ spans the full continuous domain while being parameterised by a finite number of parameters $\boldsymbol{\theta}$. In contrast to standard convolutional operations, the weights of the PCConv are not learnt directly, and are instead sampled from $g(\mathbf{e}; \boldsymbol{\theta})$. As the kernel function is itself a fully differentiable function, it can be trained through back-propagation.

In typical convolutional layers the set of output points form a subset of the input points set, provided there is no padding. However, within the parametric continuous framework, these two sets can be disjoint. This is key within the task of dMRI angular super-resolution, where the angular component of $\mathbf{y}_j$, i.e. the target b-vectors, is not contained within the set of input b-vectors.

Each PCConv layer requires three inputs to produce the output features $h_{k,j}(\mathbf{y}_j)$: the input features $f_{c,i}(\mathbf{x}_i)$, the set of input points $\mathbf{x}$, and the set of output points $\mathbf{y}$. Within each layer, following on from the right-hand side of Equation (1), the PCConv operation is defined as

$$h_{k,j}(\mathbf{y}_j) = \sum_{c=1}^{C} \sum_{i=1}^{N} f_{c,i}(\mathbf{x}_i)g_{c,k}(\mathbf{e}; \boldsymbol{\theta}). \tag{2}$$

Here, $C$ and $K$ denote the input and output feature dimensions respectively. Within each layer the number of input points $N$ is fixed, therefore the kernel function $g$ will encapsulate the $\frac{1}{N}$ factor in Equation (1) during training.

## 3.2 Convolutions in Q-Space

We define q-space $\mathbb{Q}$ as the joint spatial-angular domain within dMRI where $\mathbb{Q} = \{(u,v,w,\rho,\vartheta,\varphi)^\top \mid u,v,w,\rho \in \mathbb{R}, 0 \leq \vartheta < 2\pi, 0 \leq \varphi \leq \pi\}$ and $\mathbf{x}_j, \mathbf{y}_i \in \mathbb{Q}$. Here, $(u,v,w)^\top$ denote the spatial components, and $(\rho,\vartheta,\varphi)^\top$ denote the angular components. Specifically, $\vartheta$ and $\varphi$ refer to the azimuthal and polar angles, respectively, which form the set of points on the surface of a sphere of radius $\rho$. In this context, $\rho$ is the diffusion weighting or b-value. Figure 1 visualises

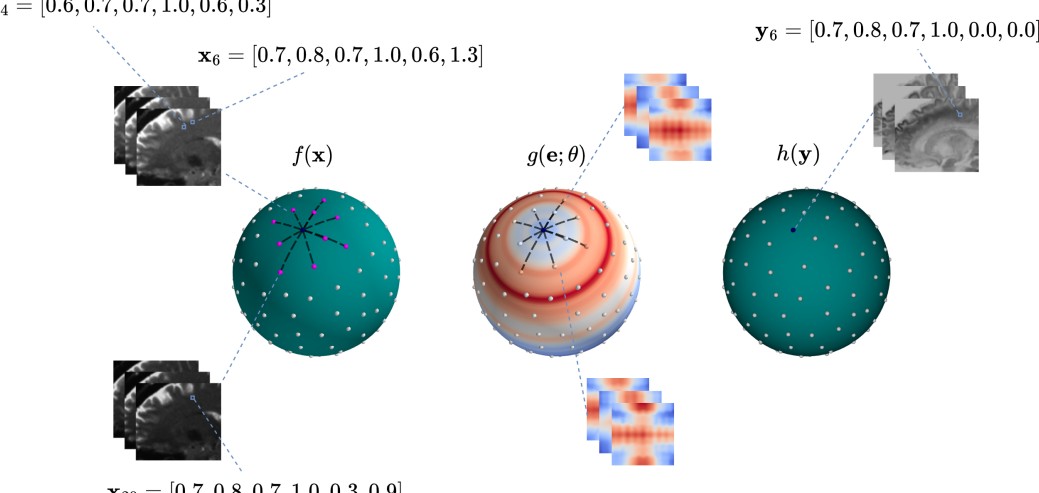

$\mathbf{x}_4 = [0.6, 0.7, 0.7, 1.0, 0.6, 0.3]$

$\mathbf{x}_6 = [0.7, 0.8, 0.7, 1.0, 0.6, 1.3]$

$\mathbf{y}_6 = [0.7, 0.8, 0.7, 1.0, 0.0, 0.0]$

$f(\mathbf{x})$

$g(\mathbf{e}; \theta)$

$h(\mathbf{y})$

$\mathbf{x}_{20} = [0.7, 0.8, 0.7, 1.0, 0.3, 0.9]$

Figure 1: Visualisation of dMRI parametric continuous convolution. Points $\mathbf{x}_i$ and $\mathbf{y}_j$ are of the form $(u, v, w, \rho, \vartheta, \varphi)^\top$ with spatial components $(u, v, w)$ and angular components $(\rho, \vartheta, \varphi)$. Left: The raw dMRI data $f(\mathbf{x})$. Points $\mathbf{x}_4$ and $\mathbf{x}_6$ are located in the same region on the q-space sphere and thus have the same angular components. Similarly, $\mathbf{x}_6$ and $\mathbf{x}_{20}$ have the same spatial coordinates but different angular components. Middle: the parametric continuous kernel, whose geometry is defined by the coordinate embedding $\mathbf{e}$. Right: Sampled points from q-space are pointwise multiplied with the kernel and summed over to produce one output feature $h(\mathbf{y}_j)$.

convolutions across q-space through Equation (2). To perform convolutions using q-space, we first calculate a coordinate embedding $\mathbf{e} \in \mathbb{R}^{2LE}$ via

$$\mathbf{e} = [\boldsymbol{\gamma}(p_1), \boldsymbol{\gamma}(p_2), \cdots, \boldsymbol{\gamma}(p_E)]^\top, \tag{3}$$

where

$$\boldsymbol{\gamma}(p_m) = [\sin(2^0 \pi p_m), \cos(2^0 \pi p_m), \cdots, \sin(2^{L-1} \pi p_m), \cos(2^{L-1} \pi p_m)]. \tag{4}$$

Here $p_m$ is the $m$th component of the coordinate vector $\mathbf{p} \in \mathbb{R}^E$ given by $\mathbf{p} = [u_i - u_j, v_i - v_j, w_i - w_j, \rho_i - \rho_j, d_r]$. We have opted to enforce rotational invariance across the sphere by omitting $\vartheta$ and $\varphi$ directly, and instead use $p_5 = d_r(\vartheta_j, \varphi_j, \vartheta_i, \varphi_i)$ where $d_r(\vartheta_j, \varphi_j, \vartheta_i, \varphi_i) = \sin(\vartheta_j)\sin(\vartheta_i) + \cos(\vartheta_j)\cos(\vartheta_i)\cos(\varphi_j - \varphi_i)$. The function $\gamma : \mathbb{R} \to \mathbb{R}^{2L}$ is a Fourier feature mapping motivated by Tancik et al. [35] and Mildenhall et al. [27] which is used to improve the hypernetwork's ability to learn high-frequency functions. In practice, $L$ is a hyperparameter of the PCConv layer. Figure 2 provides examples of weights sampled from a trained PCConv layer with varying $d_r$. The channels selected within the Figure demonstrate a variety of functions across the sphere, highlighting the network's ability to learn both high and low frequency kernels.

An important aspect of the discrete convolution is its finite support domain $M$, thus allowing the network to share parameters across the image domain. Within the PCConv framework this is achieved by limiting the number of points to convolve across, given some constraint. When convolving over q-space, the spatial dimensions follow the same constraint as in discrete convolutions. That is, for each dimension given a kernel size $n$, only the $n$ closest points are selected. As the angular components represent sparsely sampled data across a sphere, a kernel size $n$ selects the $n$ closest points, as determined by $d_r$. We refer to the angular kernel size as $k_q$ from this point onward. There is an additional constraint with the angular components that only points within a fixed radius $d_{\max}$, such that $d_r \leq d_{\max}$, are included. To achieve this, a binary mask is applied after the pointwise multiplication of the kernel weight with the input feature. The PCConv is a flexible operation that enables convolutions over non-Euclidean spaces. Additionally, the framework allows for kernel weights to be dependent upon coordinates that are *not* defined in terms of the relative distance between an input and output point. In this work, we additionally introduce a variant of the PCCNN that uses one such encoding. Denoted as 'PCCNN-Bv', this model uses $\mathbf{p} = [u_i - u_j, v_i - v_j, w_i - w_j, \rho_i, \rho_j, d_r]$.

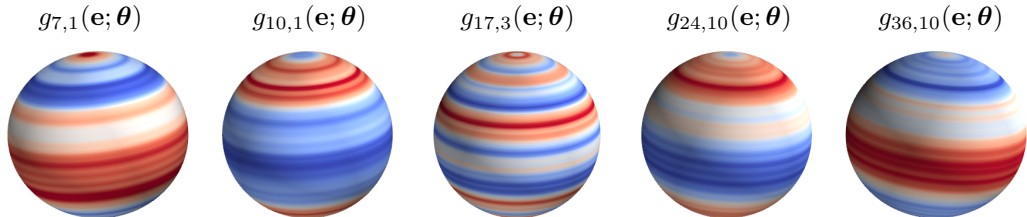

$$g_{7,1}(\mathbf{e};\boldsymbol{\theta}) \qquad g_{10,1}(\mathbf{e};\boldsymbol{\theta}) \qquad g_{17,3}(\mathbf{e};\boldsymbol{\theta}) \qquad g_{24,10}(\mathbf{e};\boldsymbol{\theta}) \qquad g_{36,10}(\mathbf{e};\boldsymbol{\theta})$$

Figure 2: Visualisation of kernel weights sampled from $g_{c,k}(\mathbf{e};\boldsymbol{\theta})$ within a PCConv layer, where $c$ and $k$ index the input and output feature dimensions respectively. To generate the spheres, $d_r$ is varied whilst all other components of $\mathbf{p}$ are kept constant.

The inclusion of this model is motivated by the non-linear relationship between different shells within multi-shell data, which may not be appropriately captured by the relative difference of b-values.

### 3.3 Global Information

Discrete convolutional layers are computed over a limited kernel size, and therefore have a limited field of view. This affords the convolutional layer computational efficiency, as the kernel size is typically much smaller than the image. However, this comes at the cost of losing global information. Within dMRI, structures within the brain are highly spatially dependent and follow a similar pattern in each individual. Non-local coordinates, and therefore global context, can be incorporated within the PCConv layer through the coordinate embedding. For example, the relative position of the kernel within an image could be incorporated into the coordinate embedding. This idea is inspired by how Vision transformers (ViTs) [12] actively extract global information through correlations between image patches. The availability of global context allows ViTs to show competitive performance in image recognition tasks, where accurate synthesis of semantic meaning more heavily relies on non-local information [19].

In clinical practice, neuroimaging data are routinely aligned to a reference space, such as Talairach space [3]. As such, coordinates within reference spaces are correlated across different subjects, and therefore can be used to provide context for the location of a voxel within the brain. In this work, after normalising the coordinates with respect to the subject brain mask size, we use the centroid of the training patches, as discussed in Section 3.6, as additional coordinates within $\mathbf{p}$. Denoted as an '-Sp' suffix, both 'PCCNN-Sp' and 'PCCNN-Bv-Sp' models within Section 4 have this modification.

### 3.4 Factorised Convolutions

dMRI is routinely stored in a dense four-dimensional format, with three spatial dimensions and one angular dimension whose coordinates are defined by $\mathbf{b}$. Performing a convolution in this high-dimensional space can be computationally expensive. We mitigate this cost by using factorised convolutions [34] within the PCConv framework. For example, in the two-dimensional case, an $(n \times m)$ kernel would be factorised into two sequential convolutions of size $(n \times 1)$ and $(1 \times m)$. As in the non-factorised case, the PCConv layer's weights are sampled from one hypernetwork $g(\mathbf{e};\boldsymbol{\theta})$. Following the approach used in Wang et al. [39], we further factorise the kernel weight tensor from $W \in \mathbb{R}^{N \times O \times C \times K}$ into two sequential operations: an input convolution $W_{\text{in}} = \mathbb{R}^{N \times O \times C}$ followed by a forward projection matrix $W_{\text{out}} = \mathbb{R}^{C \times K}$, where $O$ is the number of output points.

### 3.5 Implementation

Since PCConv layers are capable of convolving over both spatial and angular dimensions, the PCCNN models are constructed exclusively of PCConv layers and non-linear activation functions. In practice, this involves a sequence of spatially pointwise $(1 \times 1 \times 1 \times k_q)$ PCConv layers, followed by blocks of parallel convolutions consisting of spatially pointwise and $(3 \times 3 \times 3 \times k_q)$ factorised convolutions. The hyperparameters for each PCCNN, including the number of PCConv layers, can be found in Figure 5 in the Appendix. Within this work, each PCConv layer has its own independent hypernetwork. This configuration allows for significant weight sharing capabilities, as the number of parameters for each hypernetwork does not depend on the kernel size or dimensionality. Each

PCConv layer or residual PCConv block is followed by a rectified linear unit (ReLU), excluding the final layer. Each hypernetwork is composed of two dense layers, each followed by a leaky ReLU with a negative slope of $0.1$, and a final dense layer with output size 1 and no subsequent activation. The code for this work is available at github.com/m-lyon/dmri-pcconv.

### 3.6 Data and Training

Given the high dimensionality of dMRI data, using complete acquisitions as individual training examples is prohibitively expensive in terms of memory. To mitigate this issue, training sets were derived from input image patches $\mathbf{X}_{\mathrm{in}} \in \mathbb{R}^{10 \times 10 \times 10 \times q_{\mathrm{in}}}$, and target image patches $\mathbf{X}_{\mathrm{out}} \in \mathbb{R}^{10 \times 10 \times 10 \times q_{\mathrm{out}}}$ where $q_{\mathrm{in}}$ and $q_{\mathrm{out}}$ denote the number of input and target b-vectors respectively. During training, to sample the angular dimension of the input sets, an initial b-vector $\mathbf{q}_0$ was randomly selected. The next $n$, up to $q_{\mathrm{in}}$, b-vectors $\mathbf{q}$ were chosen using $\arg\max_j(\min_i(d_r(\mathbf{q}_i, \mathbf{q}_j))), \forall i, j \in \{0, 1, ..., n-1\}$. Similarly, this process was repeated to sample the output set up to size $q_{\mathrm{out}}$.

To reduce inter-subject variability, each dMRI dataset was normalised such that 99% of the data lay between $[-1, 1]$. Initially, a normalisation range of $[0, 1]$ was used; however, this resulted in training instabilities within the hypernetwork. This was likely due to a greater mismatch between the data distribution and the weight initialisation scheme within the hypernetwork, which followed $w \sim \mathcal{U}(-\sqrt{\frac{1}{k_{\mathrm{in}}}}, \sqrt{\frac{1}{k_{\mathrm{in}}}})$, where $k_{\mathrm{in}}$ was the number of input features in a given layer. dMRI data from the WU-Minn Human Connectome Project (HCP) [38] were used for training, validation and testing. This data were initially processed with the standard HCP processing pipeline [14], as well as denoised using the 'patch2self' algorithm [13]. Models were trained on twenty-seven subjects from the HCP dataset, while three subjects were used for validation during development. Hyperparameters for the PCCNN were selected through a random grid search with RayTune [20].

Each acquisition within the HCP dataset had shells of $b_{\mathrm{full}} = \{1000, 2000, 3000\}$ s/mm$^2$, each with 90 directions. Training examples contained randomly selected shells $b_{\mathrm{in}}, b_{\mathrm{out}} \sim U(b_{\mathrm{full}})$. This sampling scheme trained each network on single-shell and multi-shell examples simultaneously. The input angular dimension size was determined via $q_{\mathrm{in}} \sim U(q_{\mathrm{sample}}), q_{\mathrm{sample}} = \{6, 7, ..., 19, 20\}$. This approach allowed each network to learn the data distribution for a range of angular input sizes. To implement this, training examples had a fixed $q_{\mathrm{in}} = 20$, with zero-filled entries for input size less than 20. Models were trained using 4 NVIDIA A100's with a batch size of 16, and an $\ell_1$ loss function, for 200,000 iterations using AdamW [23]. Each PCCNN model took approximately a week to train. Details of pre-processing and training for other models can be found in the Appendix. The model weights used for test inference in Section 4 were the best performing in the validation dataset.

## 4 Experiments and Results

We evaluated the performance in dMRI angular super-resolution using eight, previously unseen subjects within the HCP dataset. Our results included four variants of the PCCNN, as listed in Table 1. As dMRI data is most commonly used for downstream analyses, we compared the results in commonly used techniques for both single-shell and multi-shell experiments. Our results included other methods where appropriate, such as spherical harmonic (SH) interpolation for single-shell experiments, FOD-Net [44] for multi-shell FBA, and SR-q-DL [41] for multi-shell NODDI. FOD-Net is a 3D CNN that maps low resolution FOD data to high resolution FOD data. SR-q-DL is a 3D CNN that maps low resolution dMRI data to high resolution NODDI data. RCNN is a 3D recurrent CNN that maps low resolution dMRI data to high resolution dMRI data. When applicable, we also compared against low angular resolution data. All error metrics were calculated with respect to the full angular resolution HCP dataset.

Table 1: Total number of parameters, in millions, of different deep learning approaches.

| Model | Parameters |
|---|---|
| SR-q-DL [41] | 0.52 |
| PCCNN | 0.77 |
| PCCNN-Bv | 0.79 |
| PCCNN-Sp | 0.83 |
| PCCNN-Bv-Sp | 0.85 |
| RCNN [26] | 6.82 |
| FOD-Net [44] | 48.17 |

The reported error values consist of the mean and standard deviation, with distribution statistics generated from the mean performance per test subject. Only voxels within the brain are included

Table 2: Absolute error of dMRI intensity in eight subjects with data derived from various models. Models use single-shell data with angular dimension size $q_{\text{in}}$ as input, and produce inferred data with angular dimension size $90 - q_{\text{in}}$ from the same shell. b-values are quoted in units of $\text{s/mm}^2$.

| | $q_{\text{in}} = 6$ | | $q_{\text{in}} = 10$ | | $q_{\text{in}} = 20$ | |
|---|---|---|---|---|---|---|
| | $b = 1000$ | $b = 3000$ | $b = 1000$ | $b = 3000$ | $b = 1000$ | $b = 3000$ |
| SH Interpolation | $84.85 \pm 5.47$ | $106.40 \pm 4.29$ | $47.25 \pm 2.85$ | $46.14 \pm 2.02$ | $45.07 \pm 3.10$ | $42.27 \pm 1.86$ |
| RCNN | $74.01 \pm 4.03$ | $\mathbf{52.11 \pm 1.97}$ | $56.01 \pm 3.50$ | $38.68 \pm 1.45$ | $53.89 \pm 3.92$ | $34.31 \pm 1.39$ |
| PCCNN | $74.55 \pm 3.69$ | $57.99 \pm 2.60$ | $48.41 \pm 2.98$ | $34.99 \pm 1.63$ | $44.03 \pm 3.09$ | $24.86 \pm 1.23$ |
| PCCNN-Bv | $\mathbf{73.55 \pm 3.63}$ | $56.71 \pm 2.56$ | $48.40 \pm 2.89$ | $34.18 \pm 1.65$ | $45.32 \pm 3.13$ | $24.77 \pm 1.20$ |
| PCCNN-Sp | $75.01 \pm 3.64$ | $57.41 \pm 2.57$ | $47.79 \pm 2.99$ | $34.16 \pm 1.49$ | $\mathbf{41.88 \pm 2.76}$ | $23.88 \pm 0.98$ |
| PCCNN-Bv-Sp | $74.87 \pm 3.51$ | $56.78 \pm 2.48$ | $\mathbf{46.92 \pm 2.53}$ | $\mathbf{33.53 \pm 1.47}$ | $42.52 \pm 2.57$ | $\mathbf{23.70 \pm 0.98}$ |

in the calculations, as determined by a brain mask segmentation. The best mean value in a given category is denoted in bold.

## 4.1 Single-shell dMRI

Within single-shell inference, we evaluated performance across two analyses. Firstly, on raw dMRI data, of which any downstream analysis task would be derived from. We compared performance across all three shells available within the HCP dataset, as well as in three q-space sub-sampling regimes, as outlined in Table 2. Secondly, we compared performance within FBA, where metrics were derived from the inferred dMRI data. For this, we considered the $b = 1000 \text{ s/mm}^2$ shell at three sub-sampling regimes.

### 4.1.1 dMRI Analysis

Table 2 presents the absolute error (AE) of the inferred dMRI data as compared to the high resolution ground truth. For the $b = 2000 \text{s/mm}^2$ results, see Table 8 within the Appendix. The PCCNN-Bv-Sp model had the lowest AE in five out of the nine shell and sub-sampling combinations and had the overall best performance when averaged across all data. Despite having approximately a tenth of the number of parameters of the RCNN [26] model, the PCCNN models had lower mean errors across all data regimes except $q_{\text{in}} = 6$ in shells $b = \{2000, 3000\} \text{ s/mm}^2$.

### 4.1.2 Fixel Analysis

FBA is an analysis method used to estimate the intra-voxel fibre populations. To generate fixel data [10], response functions were first estimated in an unsupervised manner [9], and Single-Shell 3-Tissue (SS3T) constrained spherical deconvolution (CSD) [8] was performed to obtain white matter (WM) FODs. WM FODs were then normalised [11], and finally segmented into fixels. This workflow used the MRtrix3 [37] package. FOD data is stored as a finite SH expansion, and as such, was compared using the angular correlation coefficient (ACC) [2], given by Equation (5). To compare FBA data, the AE in the apparent fibre density (AFD) within each voxel was calculated by taking the absolute difference between the magnitudes of each inferred fibre population with respect to the ground truth values.

Table 3 summarises the performance in FBA in the low resolution baseline and model based reconstructions compared to the high resolution ground truth, whilst Figure 3 visualises a subset of reconstructed FODs within an axial slice. Here, all PCCNN models performed better than the low resolution baseline in all three sampling schemes in FOD ACC, and had lower AFD AE in the two lowest sampling schemes. The PCCNN models performed competitively against the RCNN. However, in the smallest sub-sampling scheme $q_{\text{in}} = 6$, or equivalently 6.6% of the original number of voxels, the RCNN performed slightly better. Whilst the low resolution data had lower AFD AE at $q_{\text{in}} = 20$, the PCCNN models had, in the worse case, a relative increased mean error of approximately 6%, compared to 35% increase in the RCNN model. Additionally, the PCCNN models had greater ACC compared to the low resolution baseline across all sub-sampling schemes, suggesting that their use in probabilistic tractography [33] and connectomics [4] would be beneficial.

Table 3: Comparison of metrics in fixel-based analysis for eight subjects with metrics derived from both input data and inferred data using various models. Models use single-shell data ($b = 1000\text{s/mm}^2$) with angular dimension size $q_{in}$ as input, and produce $b = 1000\text{s/mm}^2$ inferred data with angular dimension size $90 - q_{in}$. *Lowres* denotes metrics derived from single-shell input only. Fibre orientation distribution (FOD) angular correlation coefficient (ACC) is defined by Equation (5), whilst apparent fibre density (AFD) absolute error (AE) is calculated using the magnitudes of all fixels within a voxel. $\uparrow$ denotes that a higher value is better.

| | $q_{in} = 6$ | | $q_{in} = 10$ | | $q_{in} = 20$ | |
| --- | --- | --- | --- | --- | --- | --- |
| | FOD ACC $\uparrow$ | AFD AE $\downarrow$ | FOD ACC $\uparrow$ | AFD AE $\downarrow$ | FOD ACC $\uparrow$ | AFD AE $\downarrow$ |
| Lowres | $0.699 \pm 0.007$ | $0.133 \pm 0.011$ | $0.793 \pm 0.009$ | $0.085 \pm 0.006$ | $0.850 \pm 0.007$ | $\mathbf{0.048 \pm 0.003}$ |
| SH Interpolation | $0.702 \pm 0.010$ | $0.180 \pm 0.014$ | $0.792 \pm 0.008$ | $0.090 \pm 0.005$ | $0.833 \pm 0.006$ | $0.078 \pm 0.004$ |
| RCNN | $\mathbf{0.708 \pm 0.005}$ | $0.081 \pm 0.006$ | $0.784 \pm 0.007$ | $0.071 \pm 0.005$ | $0.823 \pm 0.005$ | $0.065 \pm 0.004$ |
| PCCNN | $0.705 \pm 0.007$ | $0.082 \pm 0.006$ | $\mathbf{0.806 \pm 0.008}$ | $0.063 \pm 0.004$ | $\mathbf{0.856 \pm 0.006}$ | $0.049 \pm 0.003$ |
| PCCNN-Bv | $0.708 \pm 0.006$ | $\mathbf{0.079 \pm 0.005}$ | $0.802 \pm 0.006$ | $0.063 \pm 0.004$ | $0.853 \pm 0.006$ | $0.051 \pm 0.003$ |
| PCCNN-Sp | $0.699 \pm 0.007$ | $0.082 \pm 0.005$ | $0.803 \pm 0.008$ | $\mathbf{0.062 \pm 0.004}$ | $0.855 \pm 0.006$ | $0.048 \pm 0.003$ |
| PCCNN-Bv-Sp | $0.704 \pm 0.007$ | $0.081 \pm 0.005$ | $0.802 \pm 0.007$ | $0.062 \pm 0.004$ | $0.852 \pm 0.006$ | $0.049 \pm 0.003$ |

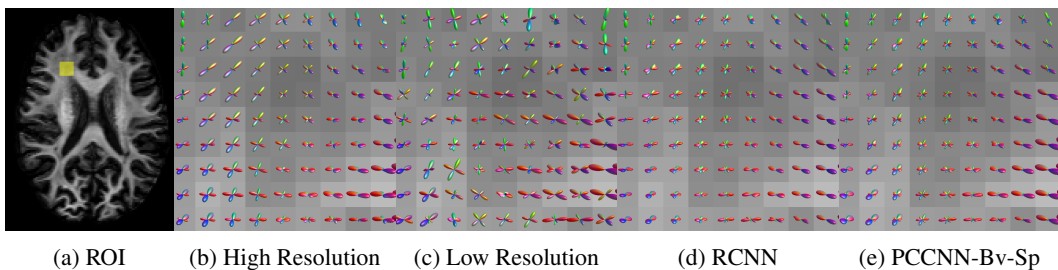

|       (a) ROI       |   (b) High Resolution   |   (c) Low Resolution   |   (d) RCNN   |   (e) PCCNN-Bv-Sp   |

Figure 3: Fibre orientation distributions (FOD)s in one ROI slice for a test subject with metrics derived from both input data and inferred data using various models. Models use single-shell data ($b = 1000\text{s/mm}^2$) with angular dimension size $q_{in} = 10$ as input, and produce $b = 1000\text{s/mm}^2$ inferred data with angular dimension size 80. Low Resolution denotes FODs derived from single-shell input only.

## 4.2 Multi-shell dMRI

To evaluate the performance of multi-shell inference, we used a varying $q_{in}$ number of $b = 1000\text{s/mm}^2$ volumes as input, and inferred the remaining known volumes from all three shells as output. The output volumes set contained the output volumes from single-shell inference, in addition to the other two shells. The $q_{in} = 6$, 10, and 20 sub-sampling schemes correspond to an under-sampling rate of 2.2%, 3.7%, and 7.4% respectively. Outlined in the sections below are the performances of the models evaluated within FBA and NODDI metrics.

### 4.2.1 Fixel Analysis

Fixel data were generated in the same manner as outlined in Section 4.1.2, except where the Multi-Shell Multi-Tissue (MSMT) algorithm [18] was used in place of SS3T. Table 4 outlines the performance within multi-shell inference. In contrast to single-shell inference, here FOD ACC and AFD AE for PCCNN models were lower than the baseline in all sub-sampling schemes. The minimum difference in AFD AE, as compared to baseline, was 13.8% at $q_{in} = 20$, whilst the maximum difference was 43.04% at $q_{in} = 6$. The FOD-Net model had the best performance overall in this experiment, obtaining the best error rate in three out of the six metrics. Despite having approximately 50-fold less parameters, the PCCNN models still perform competitively, and boasts the highest FOD ACC in $q_{in} = \{10, 20\}$.

### 4.2.2 NODDI Analysis

NODDI metrics were generated from input and inferred output volumes using the cuDIMOT [17] package. The full angular resolution derived NODDI metrics were used as ground truth. Data for $q_{in} = \{6, 20\}$ are presented in Table 5, whilst $q_{in} = 10$ can be found in Table 11 within the Appendix.

Table 4: Comparison of metrics in fixel-based analysis for eight subjects with metrics derived from both input data and inferred data using various models. Models use single-shell data ($b = 1000\mathrm{s/mm}^2$) with angular dimension size $q_{\mathrm{in}}$ as input, and produce $90 - q_{\mathrm{in}}$ $b = 1000\mathrm{s/mm}^2$, 90 $b = 2000\mathrm{s/mm}^2$, and 90 $b = 3000\mathrm{s/mm}^2$ inferred volumes. *Lowres* denotes metrics derived from single-shell input only. Fibre orientation distribution (FOD) angular correlation coefficient (ACC) is defined by Equation (5), whilst apparent fibre density (AFD) absolute error (AE) is calculated using the magnitudes of all fixels within a voxel. $\uparrow$ denotes that a higher value is better.

| | $q_{\mathrm{in}} = 6$ | | $q_{\mathrm{in}} = 10$ | | $q_{\mathrm{in}} = 20$ | |
| | FOD ACC $\uparrow$ | AFD AE $\downarrow$ | FOD ACC $\uparrow$ | AFD AE $\downarrow$ | FOD ACC $\uparrow$ | AFD AE $\downarrow$ |
|---|---|---|---|---|---|---|
| Lowres | $0.653 \pm 0.008$ | $0.157 \pm 0.014$ | $0.724 \pm 0.008$ | $0.119 \pm 0.012$ | $0.757 \pm 0.010$ | $0.086 \pm 0.011$ |
| FOD-Net | $\mathbf{0.743 \pm 0.008}$ | $0.087 \pm 0.005$ | $0.767 \pm 0.006$ | $\mathbf{0.072 \pm 0.004}$ | $0.776 \pm 0.008$ | $\mathbf{0.062 \pm 0.004}$ |
| RCNN | $0.685 \pm 0.010$ | $\mathbf{0.087 \pm 0.006}$ | $0.749 \pm 0.009$ | $0.080 \pm 0.006$ | $0.765 \pm 0.010$ | $0.079 \pm 0.006$ |
| PCCNN | $0.658 \pm 0.009$ | $0.090 \pm 0.005$ | $0.753 \pm 0.008$ | $0.077 \pm 0.005$ | $0.792 \pm 0.009$ | $0.068 \pm 0.006$ |
| PCCNN-Bv | $0.681 \pm 0.010$ | $0.091 \pm 0.005$ | $\mathbf{0.770 \pm 0.011}$ | $0.080 \pm 0.006$ | $\mathbf{0.807 \pm 0.011}$ | $0.074 \pm 0.006$ |
| PCCNN-Sp | $0.670 \pm 0.013$ | $0.092 \pm 0.006$ | $0.761 \pm 0.013$ | $0.075 \pm 0.006$ | $0.791 \pm 0.014$ | $0.070 \pm 0.007$ |
| PCCNN-Bv-Sp | $0.675 \pm 0.013$ | $0.089 \pm 0.005$ | $0.766 \pm 0.014$ | $0.075 \pm 0.006$ | $0.798 \pm 0.015$ | $0.067 \pm 0.006$ |

Table 5: Absolute error (AE) of orientation dispersion index (OD), $f_{\mathrm{intra}}$, and $f_{\mathrm{iso}}$ derived from neurite orientation dispersion and density imaging using input data and inferred data from various models. Models use single-shell data ($b = 1000\mathrm{s/mm}^2$) with angular dimension size $q_{\mathrm{in}}$ as input, and produce $90 - q_{\mathrm{in}}$ $b = 1000\mathrm{s/mm}^2$, 90 $b = 2000\mathrm{s/mm}^2$, and 90 $b = 3000\mathrm{s/mm}^2$ inferred volumes.

| | $q_{\mathrm{in}} = 6$ | | | $q_{\mathrm{in}} = 20$ | | |
| | OD | $f_{\mathrm{iso}}$ | $f_{\mathrm{intra}}$ | OD | $f_{\mathrm{iso}}$ | $f_{\mathrm{intra}}$ |
|---|---|---|---|---|---|---|
| SR-q-DL | $0.095 \pm 0.004$ | $0.162 \pm 0.015$ | $0.132 \pm 0.010$ | $0.089 \pm 0.003$ | $0.162 \pm 0.015$ | $0.129 \pm 0.010$ |
| RCNN | $\mathbf{0.074 \pm 0.007}$ | $0.030 \pm 0.003$ | $0.052 \pm 0.005$ | $0.045 \pm 0.005$ | $0.025 \pm 0.003$ | $0.048 \pm 0.006$ |
| PCCNN | $0.084 \pm 0.006$ | $0.030 \pm 0.003$ | $0.054 \pm 0.006$ | $0.039 \pm 0.006$ | $0.024 \pm 0.003$ | $\mathbf{0.045 \pm 0.004}$ |
| PCCNN-Bv | $0.075 \pm 0.006$ | $0.026 \pm 0.003$ | $0.050 \pm 0.006$ | $\mathbf{0.036 \pm 0.004}$ | $0.023 \pm 0.003$ | $0.045 \pm 0.006$ |
| PCCNN-Sp | $0.080 \pm 0.008$ | $0.026 \pm 0.004$ | $\mathbf{0.050 \pm 0.010}$ | $0.038 \pm 0.006$ | $0.025 \pm 0.004$ | $0.051 \pm 0.012$ |
| PCCNN-Bv-Sp | $0.083 \pm 0.008$ | $\mathbf{0.025 \pm 0.003}$ | $0.050 \pm 0.012$ | $0.040 \pm 0.007$ | $\mathbf{0.023 \pm 0.004}$ | $0.047 \pm 0.012$ |

Input, or low resolution, data were not included in this analysis as multi-shell data is required to reconstruct NODDI parameters. The parameters $f_{\mathrm{intra}}$ and $f_{\mathrm{iso}}$ refer to the estimated volume fraction for intra-cellular and cerebrospinal fluid respectively, whilst orientation dispersion index (OD) is an estimation of the dispersion of white matter fibre bundles.

In this experiment, the PCCNN models perform competitively, yielding the lowest error in $f_{\mathrm{iso}}$ and $f_{\mathrm{intra}}$ across all three sub-sampling rates. The relative error in OD across different models mirrors that of the FOD ACC found in Table 4, suggesting that the trained models are able to infer data that generalise across different downstream analyses. The worst performing model across all sub-sampling schemes was the SR-q-DL model, as is demonstrated qualitatively by Figure 4. Whilst this model could benefit from the more constrained task of inferring NODDI data directly, the lack of geometric prior information, such as b-vector coordinates, suggests that these additions within the PCCNN models were important to its relatively high performance in this task.

## 5 Conclusion

The results of the experiments show that the proposed PCConv operation can be used to build a flexible low parameter network to effectively infer high angular resolution raw dMRI data. It does this through appropriate incorporation of task specific geometric properties, as well as suitable parameter sharing via a hypernetwork. We show that inference of raw dMRI data can be used in downstream analysis with moderately high fidelity, and therefore demonstrates the generalisability of the model.

All three of the PCCNN models with additional modifications (PCCNN-Bv, PCCNN-Sp, PCCNN-Bv-Sp) performed greater or equal to the standard PCCNN in approximately 47% of the experiments presented in this work. Additionally, at least one of the modified PCCNNs outperformed the standard PCCNN in approximately 87% of the experiments. Overall, this suggests that the inclusion of additional domain specific context through the coordinate embedding was beneficial to the models performance.

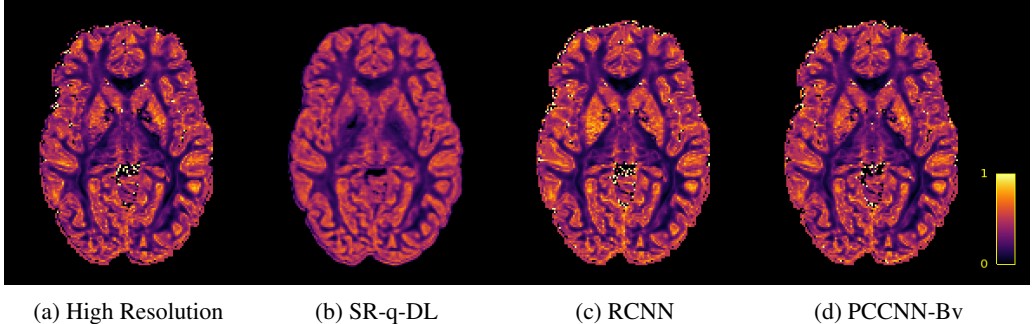

| (a) High Resolution | (b) SR-q-DL | (c) RCNN | (d) PCCNN-Bv |

Figure 4: Axial slice of orientation dispersion index (OD) within one subject across different models. Models use single-shell data ($b = 1000\text{s/mm}^2$) with angular dimension size $q_{\text{in}} = 10$ as input, and produce 80 $b = 1000\text{s/mm}^2$, 90 $b = 2000\text{s/mm}^2$, and 90 $b = 3000\text{s/mm}^2$ inferred volumes.

The application of super-resolution in a medical imaging context presents an inherent risk, as deep learning models can be susceptible to hallucinations, or high error rates, when making predictions on out-of-distribution data. Given that this work only includes data from relatively young healthy adults, future work will be needed to validate these methods in a diverse set of diagnostic settings such as datasets with pathologies, wide age ranges, and varying acquisition parameters.

Modern deep learning frameworks rely on highly optimised subroutines to perform standard discrete convolutions. Subsequently, despite having a lower number of parameters, the PCCNNs inference and train times were longer than the other methods presented in this work. This is because the PCConv uses a bespoke implementation that does not benefit from the aforementioned subroutines. This highlights the need for future research into more efficient implementations of the PCConv operation.

In summary, our work further explores the integration of geometric priors into the distinct geometry of dMRI data, and in doing so provides a flexible framework to incorporate arbitrary geometries in different domains and problem formulations.

## 6 Acknowledgements

This work was funded by the Engineering and Physical Sciences Research Council (EPSRC) Doctoral Training Partnership (DTP) Scholarship. Data were provided (in part) by the HCP, WU-Minn Consortium (Principal Investigators: David Van Essen and Kamil Ugurbil; 1U54MH091657) funded by the 16 NIH Institutes and Centers that support the NIH Blueprint for Neuroscience Research; and by the McDonnell Center for Systems Neuroscience at Washington University.

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

# A Appendix

## A.1 Model Hyperparameters

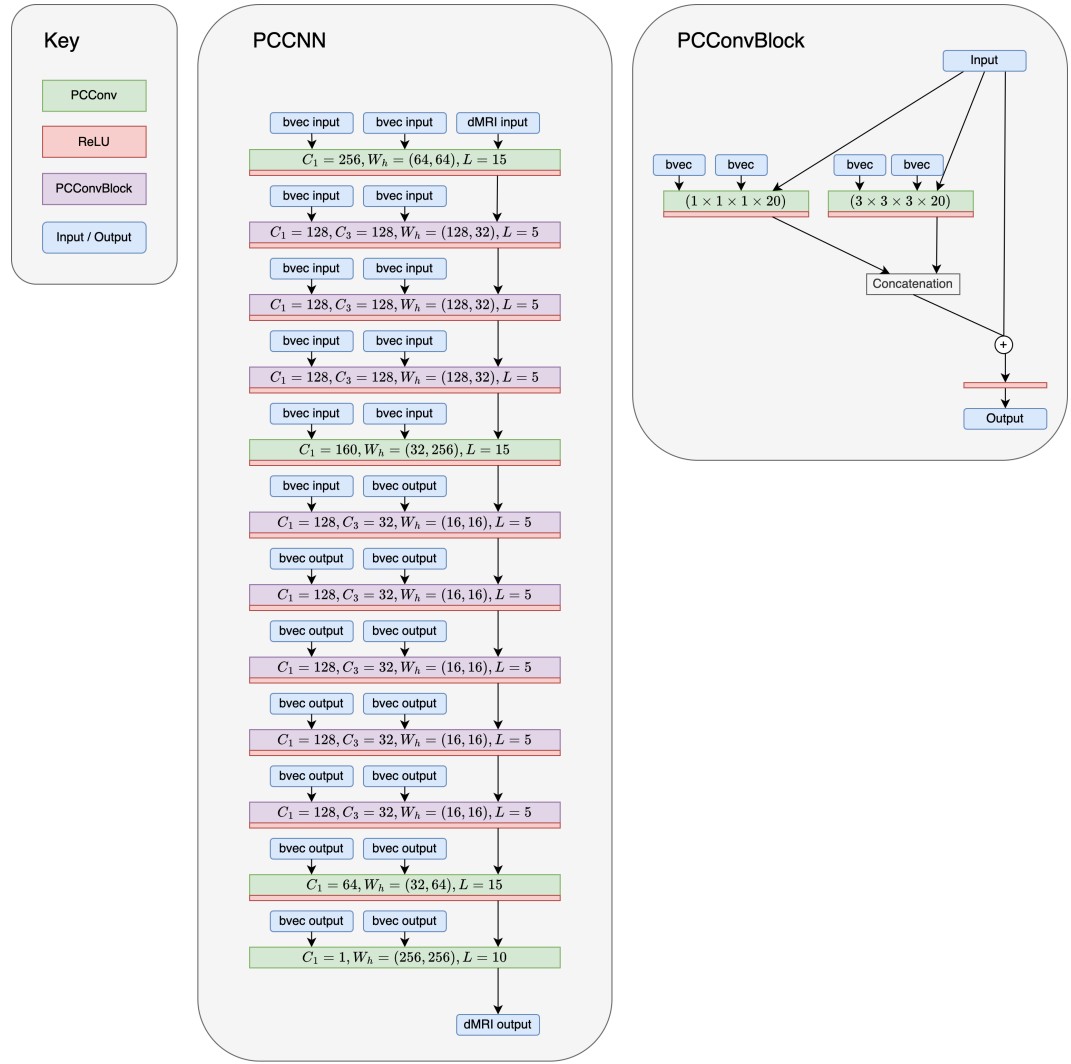

Figure 5: PCCNN model architecture. Angular kernel size is $k_q = q_{in} = 20$. $C_1$ and $C_3$ denote the number of output channels for a $(1 \times 1 \times 1 \times 20)$ and $(3 \times 3 \times 3 \times 20)$ PCConv kernel respectively. $W_h$ denotes the size of the hidden units within each PCConv's hypernetwork. $L$ is a hyperparameter of the coordinate embedding $\mathbf{e}_s$ as shown in Equation (3).

## A.2 RCNN Data and Training

Training parameters, the hardware used, and the data sampling scheme used to train the RCNN were the same as outlined within Section 3.6. Training took approximately 3 days to complete.

## A.3 FOD-Net Data and Training

In this work we modify the training scheme used within Zeng et al. [44] to more closely match the PCCNN training methodology. Specifically, each training example input was a low angular resolution FOD volume generated from single-shell dMRI data with b-value $b_{in}$ and angular dimension $q_{in}$ determined via

$$q_{in} \sim U(q_{sample}), q_{sample} = \{6, 10, 20\}, b_{in} \sim U(b_{sample}), b_{sample} = \{1000, 2000, 3000\}.$$

Table 6: Peak signal-to-noise ratio (PSNR) of dMRI intensity in eight subjects with data derived from various models. Models use single-shell data with angular dimension size $q_{\text{in}}$ as input, and produce inferred data with angular dimension size $90 - q_{\text{in}}$ from the same shell. b-values are quoted in units of $\text{s}/\text{mm}^2$.

| | $q_{\text{in}} = 6$ | | $q_{\text{in}} = 10$ | | $q_{\text{in}} = 20$ | |
| --- | --- | --- | --- | --- | --- | --- |
| | $b = 1000$ | $b = 3000$ | $b = 1000$ | $b = 3000$ | $b = 1000$ | $b = 3000$ |
| SH Interpolation | $33.83 \pm 1.48$ | $25.23 \pm 1.15$ | $39.62 \pm 1.67$ | $33.54 \pm 1.24$ | $40.04 \pm 1.65$ | $34.50 \pm 1.24$ |
| RCNN | $\mathbf{35.64 \pm 1.49}$ | $\mathbf{32.61 \pm 1.12}$ | $38.37 \pm 1.51$ | $35.80 \pm 1.15$ | $38.72 \pm 1.45$ | $37.04 \pm 1.16$ |
| PCCNN | $35.18 \pm 1.50$ | $31.13 \pm 1.15$ | $39.52 \pm 1.54$ | $36.43 \pm 1.26$ | $40.29 \pm 1.49$ | $39.93 \pm 1.40$ |
| PCCNN-Bv | $35.41 \pm 1.58$ | $31.45 \pm 1.13$ | $39.44 \pm 1.76$ | $36.77 \pm 1.27$ | $39.91 \pm 1.76$ | $39.96 \pm 1.37$ |
| PCCNN-Sp | $35.11 \pm 1.59$ | $31.25 \pm 1.15$ | $39.60 \pm 1.56$ | $36.70 \pm 1.24$ | $\mathbf{40.62 \pm 1.58}$ | $40.28 \pm 1.33$ |
| PCCNN-Bv-Sp | $35.04 \pm 1.38$ | $31.35 \pm 1.18$ | $\mathbf{39.64 \pm 1.70}$ | $\mathbf{36.93 \pm 1.23}$ | $40.27 \pm 1.66$ | $\mathbf{40.35 \pm 1.34}$ |

Table 7: Mean structural similarity index (MSSIM) of dMRI intensity in eight subjects with data derived from various models. Models use single-shell data with angular dimension size $q_{\text{in}}$ as input, and produce inferred data with angular dimension size $90 - q_{\text{in}}$ from the same shell. b-values are quoted in units of $\text{s}/\text{mm}^2$.

| | $q_{\text{in}} = 6$ | | $q_{\text{in}} = 10$ | | $q_{\text{in}} = 20$ | |
| --- | --- | --- | --- | --- | --- | --- |
| | $b = 1000$ | $b = 3000$ | $b = 1000$ | $b = 3000$ | $b = 1000$ | $b = 3000$ |
| SH Interpolation | $0.957 \pm 0.004$ | $0.788 \pm 0.010$ | $0.989 \pm 0.001$ | $0.945 \pm 0.004$ | $0.990 \pm 0.001$ | $0.954 \pm 0.004$ |
| RCNN | $\mathbf{0.973 \pm 0.002}$ | $\mathbf{0.932 \pm 0.005}$ | $0.986 \pm 0.001$ | $0.964 \pm 0.003$ | $0.988 \pm 0.001$ | $0.971 \pm 0.002$ |
| PCCNN | $0.968 \pm 0.003$ | $0.910 \pm 0.006$ | $0.989 \pm 0.001$ | $0.970 \pm 0.003$ | $0.992 \pm 0.001$ | $0.985 \pm 0.002$ |
| PCCNN-Bv | $0.970 \pm 0.003$ | $0.916 \pm 0.006$ | $0.989 \pm 0.001$ | $0.971 \pm 0.003$ | $0.991 \pm 0.001$ | $0.986 \pm 0.002$ |
| PCCNN-Sp | $0.968 \pm 0.003$ | $0.911 \pm 0.006$ | $0.989 \pm 0.001$ | $0.970 \pm 0.003$ | $\mathbf{0.992 \pm 0.001}$ | $0.986 \pm 0.001$ |
| PCCNN-Bv-Sp | $0.968 \pm 0.003$ | $0.913 \pm 0.007$ | $\mathbf{0.989 \pm 0.001}$ | $\mathbf{0.972 \pm 0.002}$ | $0.992 \pm 0.001$ | $\mathbf{0.986 \pm 0.001}$ |

Training parameters such as number of iterations, batch size, learning rate, hardware used, and optimisation method remain consistent with those detailed in Section 3.6. Model training took approximately 6 hours to complete.

### A.4  SR-q-DL Data and Training

The data sampling used to train the SR-q-DL for this work differs from that outlined in Ye et al. [41]. As in Section A.3, this was done to match the PCCNN training scheme. Here, each training example input was a low angular resolution single-shell dMRI dataset with b-value $b_{\text{in}}$ and angular dimension $q_{\text{in}}$ determined via

$$q_{\text{in}} \sim U(q_{\text{sample}}), q_{\text{sample}} = \{6, 7, ..., 19, 20\}, b_{\text{in}} \sim U(b_{\text{sample}}), b_{\text{sample}} = \{1000, 2000, 3000\}.$$

Training parameters such as number of iterations, batch size, learning rate, hardware used, and optimisation method remain consistent with those detailed in Section 3.6. Model training took approximately 30 minutes to complete.

### A.5  Spherical Harmonic Interpolation

SH interpolation of single-shell dMRI data followed the same procedure as outlined in Lyon et al. [26]. Briefly, SH coefficients for each spatial voxel within the low angular resolution dataset were fit using the pseudo-inverse least squares method using a spherical harmonic order of 2. These derived SH coefficients were then used to calculate the interpolated spatial voxels for the target b-vectors.

### A.6  Single-shell dMRI PSNR and MSSIM

Within Table 6 we present the peak signal-to-noise ratio (PSNR) of dMRI intensity within the single-shell experiment. Similarly, Table 7 outlines the mean structural similarity index (MSSIM) of dMRI intensity within the same experiment.

## A.7 Additional Single-shell dMRI Analysis

This section provides performance metrics of models in single-shell inference using $b = 2000\text{s}/\text{mm}^2$ data. Specifically, Table 8 outlines the AE of dMRI intensity, whilst Table 9 and Table 10 outline the PSNR and MSSIM of dMRI intensity respectively.

Table 8: Absolute error (AE) of dMRI intensity in eight subjects with data derived from various models. Models use single-shell $b = 2000\text{s}/\text{mm}^2$ data with angular dimension size $q_{\text{in}}$ as input, and produce inferred $b = 2000\text{s}/\text{mm}^2$ data with angular dimension size $90 - q_{\text{in}}$.

|  | $q_{\text{in}} = 6$ | $q_{\text{in}} = 10$ | $q_{\text{in}} = 20$ |
|---|---|---|---|
| SH Interpolation | $103.53 \pm 4.64$ | $45.63 \pm 2.21$ | $43.67 \pm 2.25$ |
| RCNN | $\mathbf{61.60 \pm 2.43}$ | $42.79 \pm 1.75$ | $39.14 \pm 1.93$ |
| PCCNN | $65.72 \pm 2.83$ | $37.93 \pm 1.93$ | $30.21 \pm 1.81$ |
| PCCNN-Bv | $64.63 \pm 2.74$ | $36.98 \pm 1.94$ | $30.84 \pm 1.93$ |
| PCCNN-Sp | $65.74 \pm 2.78$ | $36.82 \pm 1.77$ | $28.39 \pm 1.49$ |
| PCCNN-Bv-Sp | $65.33 \pm 2.71$ | $\mathbf{36.18 \pm 1.66}$ | $\mathbf{28.32 \pm 1.48}$ |

## A.8 Angular Correlation Coefficient

The angular correlation coefficient (ACC) is given by

$$\text{ACC}(\alpha, \beta) = \frac{\sum_{l=0}^{8} \sum_{m=-l}^{l} \alpha_{l,m} \beta_{l,m}^*}{\sqrt{\sum_{l=0}^{8} \sum_{m=-l}^{l} \alpha_{l,m}^2} \sqrt{\sum_{l=0}^{8} \sum_{m=-l}^{l} \beta_{l,m}^2}}, \tag{5}$$

where $\alpha_{l,m}$ and $\beta_{l,m}$ denote the SH coefficients for two FODs with degree $l$ and order $m$.

## A.9 Additional NODDI Analysis

Table 11 presents results from the NODDI experiment within the $q_{\text{in}} = 10$ sub-sampling scheme.

## A.10 Ablation Studies

This section describes results from various ablation studies performed with the PCCNN model as a baseline. Each of the models were used to infer multi-shell data from single-shell input data across three separate shells. The following ablations were performed:

- **No Fourier features**: The Fourier feature mapping $\gamma(p_m)$, as described in Equation (3.2) were removed from the PCConv operation.
- **No b-vectors**: The b-vector coordinates $\mathbf{e}_b$ were removed from the PCConv operation, thus reducing the operation to only convolve using spatial based co-ordinates and the b-value difference $\rho_i - \rho_j$.
- $d_{\mathbf{max}} = \frac{1}{4}\pi$: Only points on the q-space sphere at a spherical distance of $d_r \leq \frac{1}{4}\pi$ away from the centre of the kernel were included within the PCConv operation.

Table 9: Peak signal-to-noise ratio (PSNR) of dMRI intensity in eight subjects with data derived from various models. Models use single-shell $b = 2000\text{s}/\text{mm}^2$ data with angular dimension size $q_{\text{in}}$ as input, and produce inferred $b = 2000\text{s}/\text{mm}^2$ data with angular dimension size $90 - q_{\text{in}}$.

|  | $q_{\text{in}} = 6$ | $q_{\text{in}} = 10$ | $q_{\text{in}} = 20$ |
|---|---|---|---|
| SH Interpolation | $27.49 \pm 1.27$ | $35.94 \pm 1.40$ | $36.40 \pm 1.43$ |
| RCNN | $\mathbf{33.25 \pm 1.21}$ | $36.94 \pm 1.35$ | $37.79 \pm 1.36$ |
| PCCNN | $32.17 \pm 1.23$ | $37.95 \pm 1.36$ | $40.03 \pm 1.46$ |
| PCCNN-Bv | $32.43 \pm 1.23$ | $38.20 \pm 1.42$ | $39.75 \pm 1.52$ |
| PCCNN-Sp | $32.22 \pm 1.23$ | $38.22 \pm 1.38$ | $40.50 \pm 1.45$ |
| PCCNN-Bv-Sp | $32.26 \pm 1.22$ | $\mathbf{38.38 \pm 1.36}$ | $\mathbf{40.51 \pm 1.45}$ |

Table 10: Mean structural similarity index (MSSIM) of dMRI intensity in eight subjects with data derived from various models. Models use single-shell $b = 2000\mathrm{s/mm}^2$ data with angular dimension size $q_{\mathrm{in}}$ as input, and produce inferred $b = 2000\mathrm{s/mm}^2$ data with angular dimension size $90 - q_{\mathrm{in}}$.

|  | $q_{\mathrm{in}} = 6$ | $q_{\mathrm{in}} = 10$ | $q_{\mathrm{in}} = 20$ |
|---|---|---|---|
| SH Interpolation | $0.869 \pm 0.008$ | $0.972 \pm 0.003$ | $0.975 \pm 0.003$ |
| RCNN | $\mathbf{0.950 \pm 0.004}$ | $0.978 \pm 0.002$ | $0.982 \pm 0.002$ |
| PCCNN | $0.936 \pm 0.006$ | $0.983 \pm 0.002$ | $0.990 \pm 0.001$ |
| PCCNN-Bv | $0.940 \pm 0.006$ | $0.984 \pm 0.002$ | $0.990 \pm 0.001$ |
| PCCNN-Sp | $0.936 \pm 0.006$ | $0.983 \pm 0.002$ | $0.990 \pm 0.001$ |
| PCCNN-Bv-Sp | $0.937 \pm 0.006$ | $\mathbf{0.984 \pm 0.002}$ | $\mathbf{0.990 \pm 0.001}$ |

Table 11: Absolute error (AE) of orientation dispersion index (OD), $f_{\mathrm{intra}}$, and $f_{\mathrm{iso}}$ derived from neurite orientation dispersion and density imaging using both input data and inferred data from various models. Models use single-shell data ($b = 1000\mathrm{s/mm}^2$) with angular dimension size $q_{\mathrm{in}} = 10$ as input, and produce 80 $b = 1000\mathrm{s/mm}^2$, 90 $b = 2000\mathrm{s/mm}^2$, and 90 $b = 3000\mathrm{s/mm}^2$ inferred volumes.

|  | OD | $f_{\mathrm{iso}}$ | $f_{\mathrm{intra}}$ |
|---|---|---|---|
| SR-q-DL | $0.089 \pm 0.004$ | $0.162 \pm 0.015$ | $0.128 \pm 0.010$ |
| RCNN | $0.054 \pm 0.006$ | $0.026 \pm 0.003$ | $0.048 \pm 0.004$ |
| PCCNN | $0.048 \pm 0.006$ | $0.027 \pm 0.003$ | $0.049 \pm 0.006$ |
| PCCNN-Bv | $\mathbf{0.044 \pm 0.005}$ | $0.025 \pm 0.003$ | $\mathbf{0.048 \pm 0.006}$ |
| PCCNN-Sp | $0.048 \pm 0.007$ | $0.025 \pm 0.004$ | $0.048 \pm 0.010$ |
| PCCNN-Bv-Sp | $0.050 \pm 0.008$ | $\mathbf{0.024 \pm 0.004}$ | $0.049 \pm 0.012$ |

- $\mathbf{d_{max} = \frac{1}{8}\pi}$: Only points on the q-space sphere at a spherical distance of $d_r \leq \frac{1}{8}\pi$ away from the centre of the kernel were included within the PCConv operation.

Each modification was applied to all layers within the respective model. The baseline PCCNN model had a $d_{\mathrm{max}} = 1$. The results of these ablations are shown in Table 12.

Results within Table 12 clearly demonstrate that including b-vector information within the PCConv operation is essential for the model to perform well, as the AE significantly increases within the 'No b-vectors' experiment, as compared to the baseline PCCNN. The two $d_{\mathrm{max}}$ ablation results also show that removing information pertaining to the angular co-ordinate system readily reduces performance of the model. It is noteworthy that a $d_{\mathrm{max}} = \frac{1}{8}\pi$ has significantly higher AE compared to the 'No b-vectors' variant. This is likely due to the greatly reduced number of points available for each kernel operation within the $d_{\mathrm{max}} = \frac{1}{8}\pi$ experiment. This is in contrast to the 'No b-vector' variant which uses all available points within the operation, but with no angular distance information.

The inclusion of Fourier features within the PCConv operation improves performance in two of the shells, but decreases performance in one. This suggests that the inclusion of the Fourier features may improve the ability of the model to generalise to different shells, however this effect is not as apparent as the inclusion of b-vector information. The limited improvement in this case may be twofold.

Table 12: Absolute error of dMRI intensity in eight subjects with data derived from ablation models. Models use single-shell data with angular dimension size $q_{\mathrm{in}}$ as input, and infer 80 volumes within the input shell, and 90 volumes within the other two shells. b-values are quoted in units of $\mathrm{s/mm}^2$.

|  | $b = 1000$ | $b = 2000$ | $b = 3000$ |
|---|---|---|---|
| Baseline PCCNN | $51.994 \pm 2.719$ | $\mathbf{54.553 \pm 3.427}$ | $\mathbf{74.946 \pm 6.399}$ |
| No Fourier features | $\mathbf{49.681 \pm 3.386}$ | $55.619 \pm 5.190$ | $76.210 \pm 8.119$ |
| No b-vectors | $86.074 \pm 3.589$ | $89.058 \pm 5.064$ | $110.495 \pm 10.384$ |
| $d_{\mathrm{max}} = \frac{1}{4}\pi$ | $73.863 \pm 4.738$ | $85.552 \pm 9.441$ | $99.232 \pm 8.272$ |
| $d_{\mathrm{max}} = \frac{1}{8}\pi$ | $205.432 \pm 8.988$ | $222.977 \pm 13.554$ | $259.969 \pm 20.038$ |

First, due to the increased complexity of the model, as the co-ordinate embedding dimensionality would increase by a factor of $2L$. Second, whilst Fourier features were shown to vastly improve performance when predicting image intensity from co-ordinates in MLP networks [35], the Fourier feature mapping here is only applied to the co-ordinates within the domain of the kernel, which is a much smaller subset of the entire image domain.

### A.11 Noisy Data

This section provides additional results from an experiment using dMRI data that were not denoised prior to training or inference. This analysis includes an additional model, denoted the 'Q-Space CGAN' as detailed within Ren et al. [31]. The weights for this model were trained on noisy HCP data and were obtained from the original authors of the work. The other models presented in this section were trained exactly as outlined previously, with the only difference being that the data were not denoised prior to training.

Figure 6 shows an axial slice of dMRI mean absolute error (MAE) in three models when inferring multi-shell data. Whilst this is only a qualitative example, the Q-Space CGAN has significantly higher MAE across the slice as compared to the other two models. This is likely due to lack of diffusion weighted context provided to the Q-Space CGAN as input.

Table 13 presents results from the multi-shell AFD experiment, as shown in Section 4.2.1 using models trained with non-denoised data. These results follow similar trends as present in Table 4, whereby the FOD-Net generally performs best in AFD AE whilst the PCCNN variants perform best within FOD ACC across sub-sampling schemes.

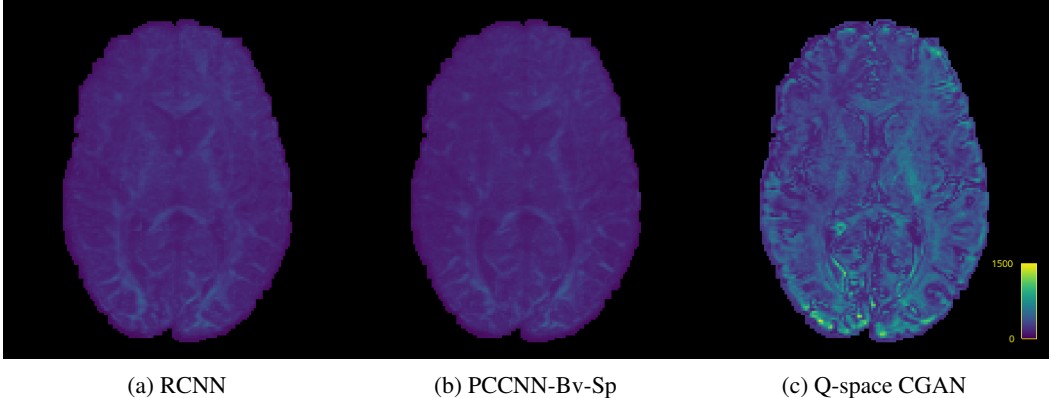

|         (a) RCNN          |        (b) PCCNN-Bv-Sp        |        (c) Q-space CGAN        |

Figure 6: Axial slice of mean absolute error (MAE) within one subject across different models. RCNN and PCCNN-Bv-Sp models use noisy single-shell data ($b = 1000\text{s/mm}^2$) with angular dimension size $q_{\text{in}} = 6$ as input, and produce 84 $b = 1000\text{s/mm}^2$, 90 $b = 2000\text{s/mm}^2$, and 90 $b = 3000\text{s/mm}^2$ inferred volumes. Q-Space CGAN uses b0, T1w, and T2w volumes to infer the same output.

Table 13: Comparison of metrics in fixel-based analysis for eight subjects with metrics derived from both noisy input data and inferred data using various models trained on noisy data. Models use single-shell data ($b = 1000\text{s/mm}^2$) with angular dimension size $q_{\text{in}}$ as input, and produce 80 $b = 1000\text{s/mm}^2$, 90 $b = 2000\text{s/mm}^2$, and 90 $b = 3000\text{s/mm}^2$ inferred volumes. *Lowres* denotes metrics derived from single-shell input only. Fibre orientation distribution (FOD) angular correlation coefficient (ACC) is defined by Equation (5), whilst apparent fibre density (AFD) absolute error (AE) is calculated using the magnitudes of all fixels within a voxel. ↑ denotes that a higher value is better.

| | $q_{\text{in}} = 6$ | | $q_{\text{in}} = 10$ | | $q_{\text{in}} = 20$ | |
| | FOD ACC ↑ | AFD AE ↓ | FOD ACC ↑ | AFD AE ↓ | FOD ACC ↑ | AFD AE ↓ |
|---|---|---|---|---|---|---|
| Lowres | $0.509 \pm 0.006$ | $0.187 \pm 0.015$ | $0.547 \pm 0.004$ | $0.145 \pm 0.011$ | $0.590 \pm 0.006$ | $0.122 \pm 0.009$ |
| FOD-Net | $0.645 \pm 0.007$ | $\mathbf{0.097 \pm 0.004}$ | $0.654 \pm 0.005$ | $\mathbf{0.084 \pm 0.003}$ | $0.669 \pm 0.007$ | $\mathbf{0.076 \pm 0.004}$ |
| Q-Space CGAN | $0.486 \pm 0.014$ | $0.160 \pm 0.013$ | $0.496 \pm 0.014$ | $0.160 \pm 0.013$ | $0.518 \pm 0.013$ | $0.156 \pm 0.012$ |
| RCNN | $0.641 \pm 0.010$ | $0.108 \pm 0.005$ | $0.707 \pm 0.017$ | $0.091 \pm 0.004$ | $0.751 \pm 0.017$ | $0.085 \pm 0.001$ |
| PCCNN-Bv-Sp | $\mathbf{0.648 \pm 0.008}$ | $0.098 \pm 0.003$ | $\mathbf{0.723 \pm 0.010}$ | $0.084 \pm 0.002$ | $\mathbf{0.762 \pm 0.011}$ | $0.078 \pm 0.001$ |

