# OpenReview forum: "Spatio-Angular Convolutions for Super-resolution in Diffusion MRI"
_NeurIPS.cc/2023/Conference — NeurIPS 2023 poster_

### Official Review · Reviewer_PJ9o · 2023-06-16

**Soundness:** 3 good
**Presentation:** 3 good
**Contribution:** 3 good
**Rating:** 6
**Confidence:** 4

**Summary:**

The authors utilize a parametric continuous convolution network to capitalize on the geometry of diffusion MRI. They enhance prior work (PCConv), integrating domain context and global information. They show that they obtain accurate high resolution dMRI using merely sparsely sampled data and demonstrate performance on two subsequent clinical tasks.

**Strengths:**

Typical CNNs do not utilize the geometry in dMRI, which opens an opportunity. The approach combines existing ideas to seize this opportunity by creating a framework that convolves over both dense grid data and sparsely sampled spherical data concurrently, augmented by context/domain information. I consider analyzing two downstream clinical tasks a considerable strength of the paper. Moreover, the paper is clearly written and provides a proper mathematical description of the method.

**Weaknesses:**

While the paper is clearly written, it is sometimes also hard to understand due to the complexity of the medical context and diffusion MRI terminology/jargon. This makes the motivation and approach of the work somewhat vague. I would suggest elaborating a bit more on the structure of the data (i.e. in paragraph 2 of the introduction) and explaining how the series, b-vector, and multi-shell are connected and what they mean/measure.
Secondly, it only becomes clear quite late that you opt for super-resolution and inferring the dMRI values from synthetically under-sampled data. I missed why this is a clinically relevant problem. Is data typically under-sampled in clinical practice? Will it speed up the acquisition? Some clinical context would be helpful.
Moreover, I feel that the results vary quite a lot throughout the experiments. Sometimes they are better than the baseline, other times they are not. While the results are as they are, I miss a proper discussion on these results to discuss why this is the case.


**Questions:**

1.	Provide more explanation on the structure of the data. Specifically, in the second paragraph of the introduction, clarify the relationship between the series, b-vector, and multi-shell, including what they measure.
2.	Make it clear earlier that the goal is super-resolution and deducing the dMRI values from synthetically under-sampled data. Explain why this is relevant in a clinical context. Is under-sampling common in practice? Would this approach speed up acquisition?
3.	Think about adding a more detailed explanation of the baseline methods. It's clear that the methods are convolutional, but how do they manage the angular data? Make it obvious what sets your work apart.
4.	Include a thorough discussion of the varying results. While it's understood that results can differ, a deeper analysis could shed light on why the outcomes are sometimes better than the baseline and sometimes not. This would enhance the readers' understanding. E.g. , why do the modifications sometimes work but not always?
5.	How does the FOD model surpasses others in the experiments discussed in section 3.2?
6.	Table 5 / Figure 3: The performance differences appear minor. Do these slight enhancements have an impact in a clinical setting? For instance, could these marginal gains influence patient treatments or clinical outcomes?


**Limitations:**

There is no limitations section present in the manuscript. A suggestion for a limitation is that the method was developed on one data set. There is, however, a very large variation between datasets, institutions, etc, in medical imaging. This also holds for MRI. It would be good to show or at least discuss the performance on other datasets or discuss whether this is a study limitation.

---

> ### Author Rebuttal · Authors · 2023-08-09
>
> # Reviewer PJ9o Rebuttal
>
> We thank the reviewer for taking the time to review our work and look forward to further discussions.
>
> ## Weaknesses
>
> We will revise the second paragraph of the introduction to further clarify the specifics of b-vectors and multi-shell acquisition. As this is an application paper in a particularly technical field it is difficult to avoid MRI terminology and jargon, however, we will revise the introduction of the camera-ready submission to increase clarity for the reader unfamiliar with dMRI terminology.
>
> The task of super-resolution is clinically relevant because of two factors: 1) MRI acquisitions in clinical settings are time-limited due to their high operating costs and desire for minimal scanning times to reduce patient discomfort. 2) Many of the downstream analysis techniques, which are prevalent in clinical *research*, require high angular resolution datasets which are typically prohibitive in normal clinical workflow due to the long scanning times they would require. Angular super-resolution looks to mitigate this problem by using models trained on prior data to produce high angular resolution datasets when only *acquiring* low angular resolution data.
>
> As a point of clarification, you mention that we are "inferring the dMRI values from synthetically under-sampled data". However, the dMRI values we infer *from*, i.e. the inputs to the model, are real (measured) data and not synthetically created.
>
> ## Questions
>
> ### Q.1 "Provide more explanation..."
>
> Diffusion MRI (dMRI) measures the degree to which water molecules can diffuse within biological tissue. This acts as a proxy for cell structure, particularly in neuroimaging, where diffusion within white matter is restricted anisotropically, whilst gray matter has isotropic diffusion. Within dMRI you can only measure the degree to which water diffuses in one spatial direction at a time. This direction is known as the b-vector and is a 3D vector on the unit sphere. To build up an appropriately informative map of the diffusion profile within the brain, you need to measure the diffusion sensitivity in many different directions. How many directions you acquire is known as the angular resolution and determines how well you can resolve certain tissue structures within the brain. You can also think of a diffusion dataset as acquiring a *series* of 3D volumes, each with a different sensitivity direction.
>
> Along with the direction of diffusion sensitivity, you can also vary the strength of the diffusion sensitivity, also known as the b-value. A stronger b-value will result in a larger contrast between an area of high diffusion (in a specific direction) and low diffusion. This is biologically relevant because different tissue (e.g. gray matter, white matter, CSF) have different, non-linear, relationships to the b-value. Therefore using multiple b-values (aka multi-shell) is key to determining the structure of the measured tissue.
>
> ### Q.2 "Make it clear earlier..."
>
> We will revise the introduction to make it more clear, early on, that the goal of this task is to infer synthetic dMRI data from under-sampled data. Under-sampling is a very common approach to angular super-resolution. This is generally the case as readily available large datasets, such as the HCP, have high angular resolution data. A typical workflow is then to select a subset of the b-vector and dMRI volume pairs and treat them as the low angular resolution input. Low angular resolution data is under-sampled (as compared to high angular resolution data) and therefore does indeed take less acquisition time.
>
> ### Q.3 "Think about adding a more..."
>
> We will revise the supplementary appendix to discuss the network architecture of the comparison methods used, making particular note of how this handles the angular component. We will also clarify how the angular components are managed briefly within the main manuscript when introducing the methods within the body of the text.
>
> ### Q.4 "Include a thorough discussion..."
>
> We agree with the reviewer that a deeper analysis would be beneficial in understanding how the modifications to the PCCNN affect performance. In particular, we will include an analysis looking at the various combinations of input shell, output shell, and input size. Training and validation loss, as well as preliminary analysis point to the fact that the PCCNN-Bv-Sp performs best for the highest number of combinations. However, why it does not *always* perform best is currently unclear. It could be the case that, due to the complexity of the problem a single model is not able to perform best in all task combinations. This can be further investigated in additional experiments by observing changes in task performance given different random weight and training initialisations, as well as observing how performance varies when models are trained on single task combinations.
>
> ### Q.5 "How does the FOD model..."
>
> The FOD-Net model likely performs well because of the constraints it imposes on the task of angular super-resolution. Specifically, it restricts the problem formulation to one downstream analysis (FOD). This imposes certain limitations on what the signal could be, as well as smoothing over high-frequency noise. Regressing over this more restricted data type is therefore arguably an easier problem, which in part explains why their model performs very well. An additional contributing factor to the FOD-Net's performance is that it only regresses over the FOD residuals. This again simplifies the problem somewhat as you only need to predict the *difference* between low-res data and high-res data. It is infeasible to predict the residuals when regressing over the raw dMRI data because the possible difference in contrast *in one voxel* between one measured direction and another is as large as the data range itself. This in effect means regressing over residuals likely would not simplify the problem.

---

> > ### Comment · Reviewer_PJ9o · 2023-08-11
> >
> > I have read the other reviews and point-by-point responses from the authors. I want to mention that the other reviewers noted points that are very important. For instance, the distinct advantages of each model modification were not evident. I appreciate the inclusion of the ablation studies. The evaluation of the performance in one test subject is still quite limited. I also feel that your statement that “why it does not always perform best is currently unclear. “ is honest, but also worrisome. This needs further exploration. Moreover, I agree that the paper is more an application paper and methodological contribution is limited.
> >
> > I have a few questions regarding my points.
> >
> > Weaknesses
> > Thanks for addressing the mentioned weaknesses, it is clearer now. I would recommend to clarify these points in the camera-ready version as well.
> >
> > Q4) “ This can be further investigated in additional experiments by observing changes in task performance given different random weight and training initialisations, as well as observing how performance varies when models are trained on single task combinations.” Is this something you are planning to do in the camera-ready version? And, are you going to include the discussion in your response to Q4 as a point in the manuscript?
> >
> > Q5) Will you discuss this in the manuscript?
> >
> > Q6) I would appreciate it if you could reflect on Q6.

---

> > > ### Author Response · Authors · 2023-08-11
> > >
> > > Thank you for taking the time to read through the reviews and comments.
> > >
> > > Whilst the evaluation over one subject is limited compared to the other analyses, given the tight timeframe we were working in it was all we were able to feasibly provide. Having said that, as mentioned within bdM9 W.1 response, one test subject still constitutes ~100,000 voxels being inferred. Additionally, given that there is a large degree of heterogeneity between patches within the brain volume, one subject still represents a reasonably wide sample of the data.
> > >
> > > Q4) Yes we will include the mentioned ablation studies and discussion into the camera-ready manuscript. We agree that the current manuscript lacks a thorough enough analysis of the different modifications to the PCCNN and so will amend it to include this.
> > >
> > > Q5) The rationale behind why the FOD-Net model uses a fundamentally a more constrained regime is important context within the analyses. We will therefore include the aforementioned discussion into the manuscript.
> > >
> > > Q6) Apologies, as we were limited by character count within the original response. See our response to Q6 and Limitations below. We have additionally added the rest of the responses to the other review threads.
> > >
> > > ### Q.6
> > >
> > > To what extent the enhancements have on real-world clinical workflows, such as patient outcomes, is an important question. Ultimately it is beyond the scope of this research, as this is establishing a methodological framework for angular super-resolution, not validating angular super-resolution within pathological workflows. However, this is certainly an area that we wish to expand the research into.
> > >
> > > To briefly comment quantitatively; if you refer to Figure 1 of the rebuttal PDF there is a visualisation of the FODs predicted from various models. Here there is a clear visual difference between the model's prediction and the low-resolution input data. This would have definitive implications for probabilistic tractography, which is typically used in studies involving [connectomics](https://www.nature.com/articles/nrn3901). These studies look at the connectivity of different regions of the brains across different populations. Therefore we would hypothesise that the application of these models would, for example, increase the validity of analyses done in connectomic studies.
> > >
> > > ## Limitations
> > >
> > > We agree with the reviewer on the need for a more comprehensive limitations section, therefore we propose the following revision:
> > >
> > > Super-resolution within a medical imaging context presents inherent risks including hallucinations or incorrect predictions, particularly when making predictions on out-of-distribution data. Given that this work only includes data from relatively young healthy adults, future work will be needed to validate these methods in a diverse set of diagnostic settings, such as datasets with pathologies, a wide age range, and a variety of acquisition parameters.
> > >
> > > Modern deep learning frameworks rely on highly optimised subroutines to perform standard discrete convolutions. Subsequently, despite having a lower number of parameters, the PCCNNs inference and train times were longer than the other methods presented in this work. This is because the PCConv uses a bespoke implementation that does not benefit from the aforementioned subroutines. This highlights the need for future research into more efficient implementations of the PCConv operation.

---

### Official Review · Reviewer_bfmq · 2023-07-04

**Soundness:** 3 good
**Presentation:** 3 good
**Contribution:** 3 good
**Rating:** 4
**Confidence:** 4

**Summary:**

The paper presents a learning-based method for q-space interpolation in diffusion MRI.  The approach constructs particular convolution operators that lend themselves to the structure of the space to be interpolated, then learn convolutions from examples.  Experiments show the learned interpolation substantially outperforms spherical harmonic interpolation and learned methods that use more generic convolutions less appropriate to the problem.

**Strengths:**

The method is well thought through and seems appropriate to the application domain.

Results are strong for the specific problem at hand.

**Weaknesses:**

Should really use a 3D q-space signal representation as a baseline, e.g. https://pubmed.ncbi.nlm.nih.gov/23587694/.

While the work is nicely formulated, it is a very niche from the point of view of a machine learning conference like NeurIPS.  I struggle to see how the methods developed have broader interest other than to researchers/users of the particular MRI modality under consideration.  The authors make no attempt to explain why this would be relevant to NeurIPS rather than an imaging conference.

**Questions:**

How does this relate to other patch-based super-resolution techniques for diffusion MRI?  Well known works on this such as https://pubmed.ncbi.nlm.nih.gov/23791914/ and image-quality transfer (https://pubmed.ncbi.nlm.nih.gov/28263925/, https://pubmed.ncbi.nlm.nih.gov/33039617/) are not mentioned at all, which is a bit worrying.

Do the technical contributions have any more general utility beyond the specific application to diffusion MRI?  Why might they be of broad interest at a machine learning conference?



**Limitations:**

Nothing to add

---

> ### Author Rebuttal · Authors · 2023-08-09
>
> # Reviewer bfmq Rebuttal
>
> We thank the reviewer for taking the time to provide this review.
>
> ## Weaknesses
>
> ### W.1 "Should really use a 3D q-space..."
>
> There are a myriad of diffusion models that we could have included in this analysis if we had unlimited time and space. For example, *dipy*, the diffusion imaging Python framework, supports [28 different](https://dipy.org/documentation/1.7.0/examples_index/#reconstruction) reconstruction models. We chose the most appropriate analysis methods based on consultation with clinical experts. Further to this point, the [two](https://arxiv.org/abs/2203.15598) other [studies](https://arxiv.org/abs/2106.13188) within the same task did not include MAP in their analysis.
>
> ### W.2 "While the work is nicely formulated..."
>
> We feel that this work is suited for publication within NeurIPS for three main reasons. Firstly, this work falls under both the "applications" and "neuroscience" categories within the NeurIPS 2023 call for papers. Secondly, there have been diffusion MRI-specific publications within NeurIPS in the past years such as [Caiafa et al.](https://papers.nips.cc/paper_files/paper/2017/hash/ccbd8ca962b80445df1f7f38c57759f0-Abstract.html), [Zheng et al.](https://papers.nips.cc/paper_files/paper/2014/hash/215a71a12769b056c3c32e7299f1c5ed-Abstract.html), [Pasternak et al.](https://papers.nips.cc/paper_files/paper/2005/hash/dfb84a11f431c62436cfb760e30a34fe-Abstract.html) and, most recently, [patch2self](https://papers.nips.cc/paper_files/paper/2020/hash/bc047286b224b7bfa73d4cb02de1238d-Abstract.html) which is included within the methodology of this study, as well as in more general [MRI studies within NeurIPS](https://papers.nips.cc/papers/search?q=MRI). Thirdly, whilst this study focuses on a specific application, it uses and extends the more general parametric continuous framework, and will provide a working implementation of the PCConv layer (of which there is none from the original paper) that can be used for any task pertaining to parametric continuous convolutions.
>
> ## Questions
>
> ### Q.1 "How does this relate to other patch-based..."
>
> Due to the brevity required of this research being published within a conference, we decided to limit the scope of the paper to angular super-resolution. [Tanno et al.](https://pubmed.ncbi.nlm.nih.gov/33039617/) is not mentioned as it only performs spatial super-resolution and therefore falls out of the scope of this page-limited publication. As further work, we intend to investigate the proposed PCConv framework in a spatial super-resolution setting, in which case this line of work will be used as a baseline. Similarly, [Coupé et al.](https://pubmed.ncbi.nlm.nih.gov/23791914/) only performs spatial super-resolution within dMRI, and additionally is arguably outdated as being published over a decade ago.
>
> Whilst [Alexander et al.](https://pubmed.ncbi.nlm.nih.gov/28263925/) represents notable work, we felt that this was out of the scope of this conference submission as it pertains to a non-deep learning-based method of regression, namely random forest and linear transform. Having said that, we feel that this and similar works are worth mentioning as context to the wider problem and will adjust the camera-ready manuscript accordingly.
>
> ### Q.2 "Do the technical contributions have any..."
>
> Broadly, this work represents the application of a general framework (parametric continuous convolutions) to a new domain (super-resolution in diffusion MRI). By demonstrating the PCConv framework we add to the body of evidence that this framework can be used in a wide array of applications. To that end, we will also provide code to apply the parametric continuous framework in general, which is of direct interest to the broader machine learning community, as it was not released with the original parametric continuous convolution publication.

---

> > ### Comment · Reviewer_bfmq · 2023-08-13
> >
> > Thanks to the authors for their responses and for considering my feedback and that of the other reviewers.
> >
> > 3D q-space interpolation: I do not agree that all 28 methods listed in dipy are equally appropriate baselines.  Ultimately the authors are trying to interpolate in q-space.  While many of the methods dipy lists under "Reconstruction" could be used that way, only a few are specifically designed for that purpose.  MAP, SHORE and other bases have clear physical motivation as choices of functions that provide appropriate continuous representations for q-space, so are a direct "physics-based" competitor for the proposed learning approach to the same problem.
> >
> > Remit. I do not really see this paper as "neuroscience", although diffusion MRI is a tool sometimes used in neuroscience.  However, the authors' response to Q2 gives me some encouragement that this work could be made more appropriate for NeurIPS.  Personally, I would have been more supportive if the work had been framed as an advance on the general topic of parametric continuous convolutions with diffusion MRI as an example application (ideally one of several).  At the very least some discussion of what other areas the proposed advances might benefit would help this work seem less niche and of broader interest.

---

> > > ### Author Response · Authors · 2023-08-14
> > >
> > > Thank you for your additional comments.
> > >
> > > ### "3D q-space interpolation: I do not agree that..."
> > > We agree that MAP, and it's 1D ~equivelant SHORE are particularly relevant reconstruction methods within dMRI. However, to our knowledge, these are multishell reconstruction methods, and therefore not applicable to this work. Specifically, the multi-shell experiments within this work are derived from single-shell data only. Therefore MAP and SHORE are not applicable as **baselines**, as they require multishell data to fit the coefficients within their respective model. MAP could be used as a downstream analysis method, similar to how FBA and NODDI are within this manuscript, however given this is a conference submission, we feel the two single-shell and two multi-hell analyses are sufficient scope for this work. We agree that a more expansive format, such as a journal submission, should include a MAP analysis.
> > >
> > >
> > > ### "Remit. I do not really see this paper as neuroscience..."
> > > Naturally as an application paper, this does not neatly fit into one category. However, we do feel strongly that this paper is an *application* paper within computer vision (line 1 of call for papers listed categories) pertaining to *health sciences* broadly (line 6) and more specifically, given diffusion MRIs wide use in both clinical and research, *neuroscience* (line 7). We feel this warrants sufficient relevance to be included within the NeurIPS proceedings.
> > >
> > > To point to a recent example, the [patch2self paper](https://arxiv.org/abs/2011.01355) published within NeurIPS 2020, is purely a dMRI application paper related to denoising dMRI data by leveraging the unique geometry of the data. We would argue this is as specific, if not more so, as our work because we relate to and extend a general (parametric continuous) framework. We will additionally provide a working implementation of the parametric continuous framework, which can be used in other domains such as [scene reconstruction](https://papers.nips.cc/paper_files/paper/2021/hash/46031b3d04dc90994ca317a7c55c4289-Abstract.html) and [neural radiance fields](https://dl.acm.org/doi/abs/10.1145/3503250), that share a similar geometric formulation.

---

### Official Review · Reviewer_Kmgz · 2023-07-08

**Soundness:** 3 good
**Presentation:** 3 good
**Contribution:** 2 fair
**Rating:** 5
**Confidence:** 4

**Summary:**

This paper proposes a parametric continuous convolution (PCConv) framework for Diffusion MRI (dMRI). The PCConv convolves across both spatial and angular dimensions of dMRI data. Meanwhile, the authors introduce a Fourier feature mapping, global coordinates, and domain speciﬁc context into PCConv. Experiments on PCCNN and three variants (PCCNN-Bv, PCCNN-Sp, PCCNNBv-Sp) show that the proposed method is competitive with less parameters.

**Strengths:**

1. Extensive experiments on both single-shell and multi-shell demonstrate that the proposed PCCNN is competitive with previous methods.
2. This paper is written clearly, and the proposed method is reasonable.

**Weaknesses:**

1. The PCCNN has been proposed in previous work [31]. The authors mainly apply the PCCNN in MRI tasks. Meanwhile, the proposed Factorised Convolutions (Section .2.5) is also designed in [31]. Therefore, the contribution and novelty of this paper are not enough.
2. The authors design four models: PCCNN, PCCNN-Bv, PCCNN-Sp, and PCCNNBv-Sp. However, in the comparisons, PCCNN-Bv-Sp combines all additions and does not achieve the best results. The performance of the four models is close to the comparison results. That is, the proposed additional improvements have little effect. The authors should demonstrate the effectiveness of the additional improvements.
3. The paper lacks an ablation study to indicate the effectiveness of each component. For example, the impacts of the angular kernel size k_q and the ﬁxed radius d_max for PCCNN. I think these are some important hyperparameters.

**Questions:**

In Line 314, the authors indicate that this method's inference and train times are longer than previous methods. Please give some explanation for this phenomenon, since the model size of PCConv is much smaller than other models.

**Limitations:**

The authors do not discuss its limitations or potential negative societal in a separate section. But some limitations (e.g., running time) are mentioned in the conclusion.

---

> ### Author Rebuttal · Authors · 2023-08-09
>
> # Reviewer Kmgz Rebuttal
>
> We thank the reviewer for taking the time to provide this review.
>
> ## Weaknesses
>
> ### W.1 "The PCCNN has been proposed in previous work..."
>
> Whilst the PCCNN developed in this study is an extension of the PCConv framework proposed by [Wang et al.](https://arxiv.org/abs/2101.06742), we feel that this work constitutes a noteworthy contribution to the literature. A significant amount of work has gone into applying the framework to the dMRI domain, such as the preprocessing steps, domain expertise in the construction of the network, choice of hyperparameters, and extension of the continuous parametric framework into higher dimensional q-space. Additionally, we provide ablation studies demonstrating optional additions to the network such as the coordinate embedding and incorporation of global information. Further, we include an extensive analysis across several different experiments to demonstrate the generalisability of our methods in different downstream dMRI workflows.
>
> ### W.2 "The authors design four models..."
>
> Given the limited rebuttal timeframe we were able to provide the following analysis to demonstrate the benefit of the PCCNN-Bv-Sp modifications. The table below looks at performance in one test subject across input shell, output shell, and input size combinations. For example, one combination would be $(b_{\mathrm{in}} = 1000, b_{\mathrm{in}} = 2000, q_{\mathrm{in}} = 6)$ and another $(b_{\mathrm{in}} = 3000, b_{\mathrm{in}} = 2000, q_{\mathrm{in}} = 10)$. Listed is the count where each model performed best in class (1st), second in class (2nd), etc. This table clearly demonstrates, in this subject, that the PCCNN-Bv-Sp performed best most of the time when looking at all combinations. However, there are some obvious caveats to this. Firstly, due to the limited timeframe we were only able to conduct this analysis within one subject. Secondly, the individual modifications do not perform better than the unmodified PCCNN. This can be further investigated in additional experiments by observing changes in task performance given different random weight and training initialisations, as well as observing how performance varies when models are trained on single task combinations.
>
> | Model       | 1st | 2nd | 3rd | 4th |
> |-------------|-----|-----|-----|-----|
> | PCCNN       | 5   | 18  | 4   | 0   |
> | PCCNN-Bv    | 4   | 3   | 16  | 4   |
> | PCCNN-Sp    | 2   | 2   | 7   | 16  |
> | PCCNN-Bv-Sp | 16  | 4   | 0   | 7   |
>
> ### W.3 "The paper lacks an ablation study..."
>
> We have included two additional ablation studies as shown by Table 2 within the rebuttal PDF. Below is a summary of those experiments.
>
> #### No B-vector Information
>
> Here we compared a PCCNN model trained up to 50,000 iterations in two scenarios: 1) with b-vectors included in the coordinate embedding "Baseline" and 2) a model with b-vectors **excluded** from the model in all layers "No b-vector". The difference in MAE, PSNR, and MSSIM all demonstrate the value of providing this information into the network. Despite only measuring data across one subject, the standard deviation within the main manuscript suggests that this difference in error is highly statistically significant. We will include a more in-depth ablation study demonstrating this within the camera-ready manuscript.
>
> #### Varying $d_{\mathrm{max}}$
>
> We compared a PCCNN model trained up to 50,000 iterations with three values of $d_{\mathrm{max}}$ set for all layers within the network: $d_{\mathrm{max}} = 1$ "Baseline", $d_{\mathrm{max}} = \frac{\pi}{4}$, and $d_{\mathrm{max}} = \frac{\pi}{8}$. Table 2 demonstrates a clear dropoff in performance across all three error metrics as $d_{\mathrm{max}}$ decreases. This is in line with what we would expect, as a lower $d_{\mathrm{max}}$ corresponds to datapoints that have a greater angular distance being excluded from the kernel. Similarly, we will include a more in-depth ablation study within the camera-ready manuscript.
>
> #### Varying $k_{\mathrm{q}}$
>
> Varying $k_{\mathrm{q}}$ would be equivalent to varying the input angular size, and is therefore demonstrated through the three input q-space sizes $(q_{\mathrm{in}} = 6, 10, 20)$ shown throughout the experiments within the main manuscript. Similarly to varying $d_{\mathrm{max}}$, this restricts the number of q-space points that is included within the kernel and effectively reduces the performance as you include fewer and fewer points.
>
> ## Questions
>
> Whilst the number of parameters within the PCCNN is much smaller than the other models presented in this work, this difference is primarily due to how the number of parameters within the PCCNN model scales with kernel size. For example, within a standard convolutional layer, the number of parameters depends on the size of the kernel, the number of input channels, and the number of output channels. Conversely, the number of parameters within the PCConv layer does not depend on the kernel size, as each weight within the kernel is sampled from the hypernetwork. Here, the PCConv layer requires more computation versus an equivalent convolutional layer due to the computation needed to sample the weights from the hypernetwork.

---

> > ### Author Response · Authors · 2023-08-11
> >
> > ## Limitations
> >
> > We agree with the reviewer's assessment that the description of the limitations should go into more detail and therefore we propose the following revision:
> >
> > Super-resolution within a medical imaging context presents inherent risks including hallucinations or incorrect predictions, particularly when making predictions on out-of-distribution data. Given that this work only includes data from relatively young healthy adults, future work will be needed to validate these methods in a diverse set of diagnostic settings, such as datasets with pathologies, a wide age range, and a variety of acquisition parameters.
> >
> > Modern deep learning frameworks rely on highly optimised subroutines to perform standard discrete convolutions. Subsequently, despite having a lower number of parameters, the PCCNNs inference and train times were longer than the other methods presented in this work. This is because the PCConv uses a bespoke implementation that does not benefit from the aforementioned subroutines. This highlights the need for future research into more efficient implementations of the PCConv operation.

---

> > > ### Comment · Reviewer_Kmgz · 2023-08-14
> > >
> > > Thanks for the rebuttal. I read the authors' responses. I also read the comments and rebuttals from other reviews. The authors provide additional experiments and analysis, which is good. However, I still have some concerns.
> > >
> > > 1. As other reviewers mentioned, it is an application paper with limited method innovations.
> > > 2. In the experiment in W2 reply, PCCNN-Bv-Sp has 11 results that are not the best, accounting for 41%. And PCCNN has 18 second best results, which is better than PCCNN-Bv and PCCNN-Sp. The results reflect that the proposed method is not effective and solid enough.

---

> > > > ### Author Response · Authors · 2023-08-14
> > > >
> > > > Thank you for your comments.
> > > >
> > > > Extending the PCCNN framework into the angular super-resolution dMRI domain required significant work and expertise. This is an application paper and therefore the point is not to establish a fundamentally new or innovative framework, but to apply and extend a previous methodology into a new domain, which is exactly what this work describes.
> > > >
> > > > The modifications to the PCCNN are ablation studies within the work and do not represent the main contribution. Instead, they serve to demonstrate the additional flexibility that the parametric continuous framework can afford. Whilst the results of the ablation studies do not outperform the non-modified model in all instances, this does not detract from the main contribution of this paper which is the application of the PCCNN to angular super-resolution in dMRI data.

---

> > > > > ### Comment · Reviewer_Kmgz · 2023-08-20
> > > > >
> > > > > Thanks for your response. I would like to increase my score to 5 (borderline accept).

---

### Official Review · Reviewer_Vyq4 · 2023-07-10

**Soundness:** 2 fair
**Presentation:** 2 fair
**Contribution:** 2 fair
**Rating:** 5
**Confidence:** 4

**Summary:**

The paper with title: Spatio-Angular Convolutions for Super-resolution in Diffusion MRI applies established fully parametric continuous convolution network (PCCNN) to diffusion SR, demonstrating the potentials.

**Strengths:**

1. This paper proposes a practical method to apply parametric continuous ConvNet to diffusion SR. Q-space in Diffusion MRI is a great example or application of continuous conv. How to leverage the diffusion gradient direction has been an open problem in the community.
2. The authors present analysis on various diffusion coefficients that are clinically useful.

**Weaknesses:**

1. The authors propose a few variations to PCCNN (i.e., BV, SP), however I did not see the clear improvements from the results what are the roles of BV and SP, they are at similar values.
2. The paper needs more comparisons to existing methods, now only RCNN in many experiments.
3. From my viewpoint, this paper misses an important ablation studies, the authors should evaluate the same architecture without the input of diffusion gradient direction and see how it contributes.
4. Needs more visual comparisons, and examples of SR results, very critical.
5. The novelty of this work is quite limited, since the core idea (PCCNN) is adopted from previous work, but application wise, its a fair application paper.



**Questions:**

1. Can this work be applied to other diffusion MRI tasks? for example diffusion MRI denoising.

**Limitations:**

1. Minor improvements over existing works.

---

> ### Author Rebuttal · Authors · 2023-08-09
>
> # Reviewer Vyq4 Rebuttal
>
> We thank the reviewer for their time in providing this review.
>
> ## Weaknesses
>
> ### W.1 "The authors propose a few variations to PCCNN..."
>
> The "Bv" modification increases the co-ordinate embedding dimensionality by splitting the coordinate $\rho_{j} - \rho_{i}$ into its constituent factors $\rho_{j}$ and $\rho_{i}$. The former coordinate only relates the *difference* between the output shell and the input shell. For example, when inferring $b = 2000$ data from $b = 1000$ data this co-ordinate would be the same as when inferring $b = 3000$ from $b = 2000$, even though the relationship between the dMRI intensity values when comparing different shells is non-linear. This is the intuition behind providing these values separately, as it allows the hypernetwork to learn this non-linear relationship explicitly.
>
> The "Sp" modification is the addition of the mean patch coordinate of the data. As each training example constitutes a small patch of the entire image, including the approximate location of the patch w.r.t. the image as a whole (i.e where the patch lies within the brain), would enable the model to modulate the kernel depending on the approximate environment the data is within. For example, patches on the exterior parts of the image would predominantly consist of gray matter, whereas inner patches would predominantly consist of white matter or csf. This difference in tissue type would in turn produce very different response functions as you vary the diffusion intensity. Overall these modifications serve to demonstrate the flexibility of the parametric continuous layer in this context. We will provide an additional analysis to further investigate the effects of the modifications for the camera-ready manuscript.
>
> ### W.2 "The paper needs more comparisons..."
>
> We have included results featuring an additional model comparison, proposed by [Ren et al.](https://arxiv.org/abs/2106.13188), within Figure 2 of the rebuttal PDF. The figure shows the mean absolute error (MAE) of two models previously included within the main manuscript (RCNN and PCCNN-Bv-Sp) as well as the model proposed by Ren et al "Q-space CGAN". The Q-space CGAN was trained with noisy data from the HCP, and given the short rebuttal timeframe we only were able to train two models with noisy HCP data, thus the limited scope of this analysis. However, we will include the Q-space CGAN in all analyses within the camera-ready manuscript.
>
> Within these preliminary results, Figure 2 clearly shows a distinctly higher MAE in the Q-space CGAN as compared to the two other models. This suggests a similar trend will be present when applying this method to other experiments. In addition to this, as stated in the response to Reviewer bdM9, we will endeavor to compare our methods against an equivariant dMRI network (see section C.1 of bdM9 response), as well as dMRI analysis coefficient regression methods (see section C.3 of bdM9 response) within the camera-ready manuscript.
>
> ### W.3 "From my viewpoint, this paper misses an important ablation..."
>
> We agree with the reviewer's concern for more ablation studies and have conducted the following experiments, as shown in Table 2 within the rebuttal PDF.
>
> #### No B-vector Information
>
> Here we compared a PCCNN model trained up to 50,000 iterations in two scenarios: 1) with b-vectors included in the coordinate embedding "Baseline" and 2) a model with b-vectors **excluded** from the model in all layers "No b-vector". The difference in MAE, PSNR, and MSSIM all demonstrate the value of providing this information into the network. Despite only measuring data across one subject, the standard deviation within the main manuscript suggests that this difference in error is highly statistically significant. We will include a more in-depth ablation study demonstrating this within the camera-ready manuscript.
>
> #### Varying $d_{\mathrm{max}}$
>
> We compared a PCCNN model trained up to 50,000 iterations with three values of $d_{\mathrm{max}}$ set for all layers within the network: $d_{\mathrm{max}} = 1$ "Baseline", $d_{\mathrm{max}} = \frac{\pi}{4}$, and $d_{\mathrm{max}} = \frac{\pi}{8}$. Table 2 demonstrates a clear dropoff in performance across all three error metrics as $d_{\mathrm{max}}$ decreases. This is in line with what we would expect, as a lower $d_{\mathrm{max}}$ corresponds to datapoints that have a greater angular distance being excluded from the kernel. Similarly, we will include a more in-depth ablation study within the camera-ready manuscript.
>
> ### W.4 "Needs more visual comparisons..."
>
> We have provided two new figures within the rebuttal PDF. Figure 1 shows visualisations of the fibre orientation distribution (FOD) maps in a crossing fibre region within one subject, across various models. Here it can be seen that the angular super-resolution models improve the qualitative fidelity of the FODs as compared to the high-resolution baseline. This visualisation helps to demonstrate the value that these super-resolution models would have for applications such as tractography, which uses FODs to determine streamlines.
>
> Figure 2 is discussed in comment W.2 of this response and pertains to the MAE in an additional model, the "Q-space CGAN". We will expand this figure within the camera-ready manuscript to include more models and are happy to include other figures and visualisations in addition to this.

---

> > ### Author Response · Authors · 2023-08-11
> >
> > ### W.5 "The novelty of this work is quite limited..."
> >
> > Whilst the PCCNN developed in this study is an extension of the PCConv framework proposed by [Wang et al.](https://arxiv.org/abs/2101.06742), we feel that this work constitutes a noteworthy contribution to the literature. A significant amount of work has gone into applying the framework to the dMRI domain, such as the preprocessing steps, domain expertise in the construction of the network, choice of hyperparameters, and extension of the continuous parametric framework into higher dimensional q-space. Additionally, we provide ablation studies demonstrating optional additions to the network such as the coordinate embedding and incorporation of global information. Further, we include an extensive analysis across several different experiments to demonstrate the generalisability of our methods in different downstream dMRI workflows.
> >
> > ## Questions
> >
> > ### Q.1 "Can this work be applied to other diffusion MRI tasks?..."
> >
> > Yes, this absolutely can be applied to other diffusion MRI tasks. For example, the network could be used as-is on a denoising task. The only difference would be the input and target data, as they would need to be replaced with noisy and denoised data respectively. For other tasks such as segmentation or classification, modifications to the network hyperparameters (such as the dimensionality of layers) may need adjusting to accommodate these new tasks, but the PCConv building blocks would remain the same.

---

> > > ### Comment · Reviewer_Vyq4 · 2023-08-11
> > >
> > > Thanks for the responses.
> > > I'm happy to increase my score.

---

### Official Review · Reviewer_bdM9 · 2023-07-23

**Soundness:** 2 fair
**Presentation:** 4 excellent
**Contribution:** 2 fair
**Rating:** 5
**Confidence:** 5

**Summary:**

**Background for ML audiences**: Multi-shell diffusion MRI (dMRI) is a 6D (3 dims of space + 2 angular dimensions + 1 radial dimension) imaging modality where each 3D voxel contains (potentially concentric) spherical signals which correlate with local white matter properties within the brain. In research settings, each voxel-sphere is sampled densely to recover underlying white matter pathways, but clinical settings often acquire only a few angles for speed in critical settings (e.g. stroke monitoring).

**Summary**: Submission 13443 presents an architecture for processing dMRI images and aims to perform angular super-resolution. To do so, it applies existing parametric continuous convolutional nets (i.e. continuous (x,y,z,r,phi,theta)-location conditioned networks) and adds better positional embeddings and positional conditioning. Experimentally, it evaluates angular super-resolution performance by means of plain predicted image-based performance and downstream analysis such as NODDI parameter fitting.

**Strengths:**

- To my knowledge, fully continuous convolutions for dMRI processing has not been attempted before and the idea is promising due to the extremely nonlinear nature of the 5D spatial and angular images.
- The submission performs a reasonably thorough ablation of its various moving components.
- The analysis of downstream applications of dMRI (fixel analysis, NODDI fitting) is well considered as an evaluation strategy of angular super-resolution.

**Weaknesses:**

### (A) Inadequate evaluation with lower methodological contributions:

The proposed framework is an application of continuous convolutions to dMRI images alongside combinations with factorized convolutions and Fourier features, the latter of which has been used extensively with coordinate-based hypernetworks and equivalently for [continuous convolutions](https://arxiv.org/abs/2102.02611). In my opinion, this is fine as dMRI processing is a non-trivial and non-traditional application. However, the presented experiments require significant improvements in depth to make this argument. For example,

- Only ~**40 subjects** were used for training/validation/testing with final evaluation only performed on 8 subjects. However, the publicly and freely accessible dataset used in this paper (WU-Minn HCP) has diffusion MRI from 1065 subjects. Why was the entire dataset (or at least a much larger portion) not used? In my opinion, this is a non-trivial limitation as this sample size is small and the evaluation on 8 subjects only reveals no clear results trend amongst all of the various ablations proposed.
- The paper claims to improve PCCNN networks in performance and computational efficiency with the addition of factorized convolutions and Fourier feature embeddings. However, the gains specific to factorized convolutions and FF embeddings do not appear to be quantified in an **ablation study**, please do so as they are central claims.
- There is no mention of how the **baselines** were tuned for these experiments. For example, was the degree of the SH interpolation tuned on validation data?

### (B) Potentially suboptimal modeling choices:

- Section 2.4 introduces “**global conditioning**” by means of concatenating coordinates of a reference neuroimaging template. However, this is quite confusing as dMRI are routinely *not* registered to a template space as that requires non-trivial reorientation of each voxel-sphere after registration, so I do not see why this would be helpful and the ablation results are inconsistent and hard to interpret. Further, it is unclear what is technically meant by L159. Please clarify these aspects.
- It is unclear and unstated why the proposed framework uses **rotationally-invariant kernels**. The main advantage of the proposed work over previous attempts (listed below) is that the hypernetwork can be quite expressive for highly nonlinear data, but rotationally-invariant kernels reduce this to a significant extent. Why was this choice made?
- The paper performs **zero-filling** of missing angles in its experiments (L208) which in my opinion is not well motivated as 0 diffusivity is a physical value and not a masked missing value to be estimated. As continuous convolutions naturally allow for handling missing values, I do not see why zero-filling is necessary, please clarify.
- All experiments undergo **denoising** via Patch2Self which would typically significantly reduce the high-frequency content of the sparsely-sampled image which could potentially be useful for nonlinear neural networks. Why is this preprocessing used?
- As the proposed method is not roto-translation equivariant by design (as in other dMRI processing networks), it is largely data-driven. However, only 27 training images are used and there is no mention of data augmentation which would help with generalization. Was **data augmentation** not used?

### (C) Missing existing work on this topic:

Currently, most experiments in the paper use 1-2 baseline methods as comparisons, which would be fine if the submission was tackling a recently-formed application. However, there is extensive work on topics relevant to the submission such as dMRI convolutions, angular super-resolution, and regressing microstructural coefficients which are detailed below. It would improve the paper for it to contextualize its contributions w.r.t. existing work and (if possible) add one or more of the most relevant baselines to each experiment.

#### w.r.t. dMRI convolution and/or q-space processing:
The current version of this paper presents convolutions for dMRI data as a fundamentally new and unaddressed problem. However, there are several lines of existing work that tackle developing convolutions for dMRI exactly. This includes equivariant and geometrically-motivated work:
- Using tensor field networks: https://arxiv.org/abs/2102.06942
- Using separable kernels: https://openreview.net/forum?id=7S1l2zzUZFI
- Using equivariant graph convolutions: https://arxiv.org/abs/2304.06103
- Using manifold-valued convolutions: https://link.springer.com/chapter/10.1007/978-3-030-78191-0_24
- And for completeness, there exist ad-hoc methods which concatenate neighboring voxel-spheres channel-wise such as https://hal.science/hal-02946371/document.

In my opinion, it would be good to acknowledge and discuss the advantages and disadvantages of the proposed framework w.r.t. one or more of these methods. For example,
- A potential disadvantage: the proposed method is not equivariant to the underlying symmetries of the data and is purely data-driven.
- A potential advantage: the above methods cannot directly produce outputs at angular coordinates not in the inputs without interpolation and/or model-fits, whereas the proposed method can as it uses a hypernetwork.

#### w.r.t. dMRI super-resolution

DMRI angular super-resolution has had several previous attempts in the literature that could be discussed and/or compared against when possible. For example, [DeepDTI](https://www.sciencedirect.com/science/article/pii/S1053811920305036) uses 6 angular samples (w/ universally co-acquired structural images) to super-resolve the diffusion tensor. Further, while the paper uses continuous convolutions via hypernetworks both spatially and spherically, the continuous mechanism is not used spatially as all inputs and outputs lie on a regular spatial grid. As a result, it is similar to [this paper](https://arxiv.org/abs/2106.13188) which also uses a q-space-conditioned hypernetwork with spatial gridded convolutions for dMRI super-resolution. Lastly, while SH interpolation is compared against as a non-deep learning baseline, one would typically use SHORE interpolation in practice.

#### w.r.t. regressing dMRI coefficients:

There are some works that directly regress dMRI scalars (such as NODDI in this paper) which would also be good to discuss and/or compare against if possible such as:
- https://onlinelibrary.wiley.com/doi/full/10.1002/mrm.27568
- https://link.springer.com/chapter/10.1007/978-3-031-16431-6_15
- https://arxiv.org/abs/2207.00572
- https://link.springer.com/chapter/10.1007/978-3-031-16431-6_11

### Minor comments
- Why is the related work in the methods section? That choice seems out of place.
- Please add PSNR and SSIM as evaluation measures on top of AE.

**Questions:**

I have merged my questions and suggestions with the weaknesses (mainly weaknesses A and B), please see above. If the rebuttal adequately addresses weaknesses A and B I would be happy to raise my score.

**Limitations:**

The paper does not list any limitations beyond long compute times. In my opinion, this is insufficient as angular super-resolution in biomedical applications is a high-risk application as it may introduce hallucinations or incorrectly predict angular samples for unseen populations. Further, the paper only studies 40 “normal” relatively-young adults, whereas angular super-resolution would be most beneficial in time-sensitive clinical settings with lesions and strokes. These limitations should be addressed.

---

> ### Author Rebuttal · Authors · 2023-08-09
>
> # Reviewer bdM9 Rebuttal
>
> Thank you for taking the time to provide such a robust and thorough review, it is greatly appreciated.
>
> ## Weaknesses
>
> ### A.1 "Only ~40 subjects were used for training/validation/testing..."
>
> Whilst only ~40 subjects were used in total, each subject constitutes a lot of data. For example during training, each subject is split up into patches of dimensions $(10, 10, 10, q_{\mathrm{in}} + q_{\mathrm{out}})$ where $q_{\mathrm{in}}$ and $q_{\mathrm{out}}$ form the input and target set respectively. Overall, each of the 27 training images provided on the order of ~4000 unique training examples, or ~100,000 across all subjects. In addition to this, each training example will contain on the order of 20,000 voxels. The evaluation statistics for the 8 test subjects span ~600,000,000 voxels in total, depending on the experiment. Given that each datapoint within each table in the manuscript is computed from 600M voxels, increasing the test dataset size would drastically increase the test evaluation computation time. Similarly, increasing the training dataset size to the order of hundreds of subjects would dramatically increase training time.
>
> Other deep learning-based studies have used a similar number of subjects from the HCP, for example; [Ren et al.](https://arxiv.org/abs/2106.13188) used 9 for training and 9 for test, [Alexander et al](https://www.sciencedirect.com/science/article/pii/S1053811917302008) used 8 for training and 8 for test, and [Ye et al](https://link.springer.com/chapter/10.1007/978-3-030-32248-9_65) used 5 for training and 20 for test.
>
> ### A.2 "The paper claims to improve PCCNN networks..."
>
> Performing an ablation study by removing the factorised convolutions would be technically difficult due to the memory limit implications it would pose. Specifically, the current model(s) would require too much memory to fit into our available GPU hardware, and therefore a significantly smaller model would be required to be able to test this. The high memory requirements of the model are due to the relatively high dimensionality of the data, and therefore kernels, involved. For example, a typical 2D natural image convolutional kernel would involve the multiplication of $(3 \times 3) = 9$ weights per input channel per output channel. Conversely, our non-pointwise non-factorised kernel would use $(3 \times 3 \times 3 \times 20) = 540$ weights per input channel per output channel.
>
> Whilst we did not have time to conduct the ablation study that detailed the effects of removing the Fourier features, we were able to run small-scale ablations asked by reviewers Kmgz and Vyq4. These ablations look at removing b-vector information and varying $d_{\mathrm{max}}$ within the PCCNN network. We refer the reader to the rebuttals of these two reviewers for a more detailed discussion of those results. We will of course provide an ablation study of the Fourier features for the camera-ready submission.
>
> ### A.3 "There is no mention of how the baselines were tuned..."
>
> Details, such as data and training regime, for the FOD-Net and SR-q-DL model comparisons were listed in the Appendix and referred to on line 211. Details of the RCNN model comparison were omitted as they did not differ from the PCCNN models. However, we will amend the Appendix to explicitly state this for clarity. We will also include the spherical harmonic interpolation procedure, and amend the manuscript to mention this. See below for an example of the amendment we will make.
>
> #### RCNN Data and Training
>
> Training parameters, the hardware used, and the data sampling scheme used to train the RCNN were the same as outlined within Section 2.7. Model training took approximately 3 days.
>
> #### Spherical Harmonic Interpolation
>
> Spherical harmonic interpolation of single-shell dMRI data followed the same procedure as outlined in [Lyon et al.](https://arxiv.org/abs/2203.15598). Briefly, SH coefficients for each spatial voxel within the low angular resolution dataset were fit using the pseudo-inverse least squares method using a spherical harmonic order of 2. These derived SH coefficients were then used to calculate the interpolated spatial voxels for the target b-vectors.
>
> ### B.1 "Section 2.4 introduces global conditioning...”
>
> Indeed dMRI are not routinely registered to template space due to the non-trivial reorientation. However, only the affine transformation, which relates voxel space to template space, is required to obtain the coordinates of patches within template space. Therefore a workflow would look like this: 1) register your dMRI to a dMRI template space. 2) Throw away the registered dMRI data whilst keeping its affine transformation. 3) Use that affine transformation to calculate the template space coordinates.
>
> Including this coordinate is hypothesised to be helpful as it provides the model with extra context for where the input (patch) data is located within the whole image/brain, similar to how ViT's encode patch order via positional encoding.
>
> L159 describes how the patch-wise spatial coordinate is normalised to a range $[0, 1]$, via the brain mask. For example, a patch located in the inferior region of the brain would have a z-component of close to 0.
>
> ### B.2 "It is unclear and unstated why the proposed framework..."
>
> In preliminary experiments, we found that restricting the kernel to be rotationally invariant yielded better training results and therefore included this in the final model(s). Whilst we did not have time within the review period to do so, we will provide an ablation study within the camera-ready manuscript to demonstrate these results.

---

> > ### Author Response · Authors · 2023-08-11
> >
> > ### B.3 "The paper performs zero-filling..."
> >
> > Zero-filling is necessary from an implementation standpoint because, while the continuous framework naturally handles missing values, the data are stored and computed with dense tensors. Specifically, during training, each batch will have training examples with various non-empty angular dimension sizes contained within one dense tensor. The minimum size of the tensor in this dimension should be the largest angular dimension size possible and have zero filled values where data are not present. Importantly though, this is not equivalent to treating the data as b0 volumes. Firstly, the zero-filled values are masked such that none of these empty values contributes to the gradient update. Secondly, whilst b0 volumes are not included in this study, their inclusion would be marked by the $\rho_{i}$ or $\rho_{j}$ component within the $\mathbf{p}$ vector.
> >
> > ### B.4 "All experiments undergo denoising..."
> >
> > Denoising is an important step in the diffusion processing pipeline, and given that the method demonstrated superior denoising capabilities compared to others (as demonstrated within the patch2self [paper](https://arxiv.org/abs/2011.01355)), we opted to use it as part of the preprocessing for this study.
> >
> > Additionally, the formulation of patch2self lends quite naturally to angular super-resolution. In essence, patch2self removes noise from a given 3D dMRI volume $v_{j} \in \{v_{1}, \ldots, v_{N} \}$ by posing it as a prediction problem. Linear regression is used to predict $v_{j}$ by using all other $N -1$ volumes as input. This process removes noise by using regression coefficients that are calculated via the averaging across all voxels within a given 3D dMRI input volume $v_{j}$. Assuming noise is uncorrelated, the best prediction an angular super-resolution model could yield would be equal to this mean value predicted by the denoiser.
> >
> > Having said that, patch2self, or indeed any denoising step, is not *necessary* in this application and the model(s) could be trained with noisy data. Indeed we have the RCNN and PCCNN-Bv-Sp models trained on noisy data, and results from which are included in Figure 2 of the rebuttal PDF. Whilst we hypothesise the trends across models and experiments will be the same, we will include an ablation experiment within the camera-ready submission demonstrating this.
> >
> > ### B.5 "As the proposed method is not roto-translation equivariant..."
> >
> > Whilst only 27 training images are used, as stated previously, the training dataset constitutes approximately 100,000 unique training examples. This was deemed a reasonable amount of data to use and is concurrent with other similar studies. Additionally, augmentation was not used due to the added complexity that this would yield when dealing with dMRI data. Specifically, as you previously pointed out, deformations are not routinely used in dMRI processing due to the non-trivial effects they would impose on each voxel-sphere. Therefore it is not immediately clear what a valid augmentation to the data would, and would not, be whilst maintaining a realistic dataset.

---

> > > ### Author Response · Authors · 2023-08-11
> > >
> > > ### C.1 "w.r.t. dMRI convolution and/or q-space processing"
> > >
> > > It was not our intention to present convolutions for dMRI as a fundamentally new problem. As you have highlighted there has been excellent work done within the geometric deep learning field on incorporating the mathematical frameworks of equivariant networks into dMRI deep learning. Whilst, to our knowledge, there are no works that use the equivariant networks within angular super-resolution, it would nonetheless be a good choice to develop a network comprised of such layers to serve as a comparison to our method. Good candidates for this would be derived from either [Muller et al.](https://arxiv.org/abs/2102.06942) or [Elaldi et al.](https://arxiv.org/abs/2304.06103). Whilst we are unable to produce this during the rebuttal timeframe, this is something we will do for the camera-ready submission.
> > >
> > > In the meantime, we can certainly speak to some advantages and disadvantages of the equivariant methods.
> > >
> > > Regarding advantages, these equivariant dMRI frameworks lend naturally to the geometry present within the data and enforce appropriate symmetries, such as rotational and translational, that guarantee validity given transforms within that domain. Additionally, because of these equivariances, roto-translational transformations are explicitly tied into the model and therefore do not need to be learnt via examples or augmentation within the dataset. This generally allows for these methods to be more appropriate in low-data domains.
> > >
> > > As you pointed out, the formulations do not come with an out-of-the-box way to produce outputs at specific angular coordinates. In this respect, there is less flexibility regarding coordinate system choice. On that note, the continuous convolution framework is much less rigid when introducing additional coordinate systems. Equivariant networks are limited regarding network choices, as certain non-linearities break the guarantee of equivariance, whereas our method can be used in conjunction with any non-linearity or sequence of layers.

---

> > > > ### Author Response · Authors · 2023-08-11
> > > >
> > > > ### C.2 "w.r.t. dMRI super-resolution"
> > > >
> > > > Within previous experiments using DTI data derived from angular super-resolution methods, we found that the quantitative gains in error compared to a ground truth from using these methods were minimal in DTI metrics. We attribute this is due to the relative simplicity of the DTI framework. For example, the DTI tensor cannot resolve different fibre populations within one voxel. Additionally, DTI only requires 6 diffusion directions to fit the diffusion tensor, compared to a recommended ~30 directions to fit [FODs](https://pubmed.ncbi.nlm.nih.gov/27639350/). [Lyon et al](https://arxiv.org/abs/2203.15598) demonstrated that what gains you do get from angular super-resolution are not greater than the increased uncertainty you introduce from using these deep learning models, in q-space sampling schemes for ten and above. This, and in the interest of brevity, is our rationale for omitting DTI from this study.
> > > >
> > > > The method presented by [Tian et al.](https://www.sciencedirect.com/science/article/pii/S1053811920305036) requires both T1 and T2 structural scans as part of their core methodology, whereas ours requires no structural data. Additionally, this method only yields an additional 6 spatio-angular volumes, only infers single-shell data, and lacks a way to sample arbitrary angular directions at test time.
> > > >
> > > > [Ren et al.](https://arxiv.org/abs/2106.13188) is indeed similar work in this task and is worth inclusion in this study. There are a few differences worth noting between the two models, however. Firstly, whilst they do employ multiple layers to condition the generator network on q-space information, these layers, we would argue, are not conceptually the same as a hypernetwork, as weights are not sampled from these conditioning layers, and instead intermediate layers are modulated via this conditioning network. Secondly, this work uses structural T1w, T2w and b0 images as input. This, in our opinion, is potentially a nonoptimal choice for two reasons. Firstly because b0 images follow the same contrast as T2w images, there would be a high amount of redundant information when using both. Secondly, there is no diffusion-weighted information in this setup. This means that at test time, any directionally sensitive contrast is derived entirely from the training data. This would not only likely lead to higher reconstruction error, but any signal not covered by the training data would be highly likely to hallucinate features.
> > > >
> > > > The code and pre-trained weights were available for their q-space CGAN network, and therefore we were able to produce some preliminary results that corroborated our hypothesis. Figure 2 shows the mean absolute error (MAE) within an axial slice in one test subject. In this experiment, all three models were trained on noisy HCP data. Figure 2 clearly shows a stark difference in MAE within the Q-space CGAN as compared with the RCNN and PCCNN-Bv-Sp models. Whilst this is a preliminary analysis, we will include this method in the full set of experiments in the camera-ready manuscript.
> > > >
> > > > Whilst we were unable to incorporate SHORE into our experiments during the rebuttal timeframe, we will include SHORE as a baseline within the final manuscript.
> > > >
> > > > ### C.3 "w.r.t. regressing dMRI coefficients"
> > > >
> > > > Regressing over dMRI-derived coefficients is a reasonable approach to the problem of angular super-resolution, however, two points are worth emphasising. Firstly, there are many different dMRI analysis techniques and limiting regression over one analysis limits the generalisability of the model. This was a strong motivating factor as to why we chose this formulation. Secondly, the angular relationship between the data is often expressed in a compressed manner, i.e. each analysis technique enforces assumptions which compress the angular dimension into a scalar quantity. Therefore regressing over these coefficients does not require explicit handling of the angular geometry, therefore simplifying the task. Nonetheless, it is an important baseline to compare against as each of these models is a competitor in their respective analysis framework (whether that be NODDI, FOD, etc). The works you highlighted represent important work within the field, and we will include additional models for the camera-ready manuscript. Whilst an initial search yielded no implementation or code for the four studies, we will endeavour to include as many comparisons as possible given the timeframe.
> > > >
> > > > ## Minor Comments
> > > >
> > > > ### M.1 "Why is the related work in the methods section?..."
> > > >
> > > > Initially, we thought it natural to include related work within the methods section as that section goes into some detail about downstream analysis methods used within the experimental section, such as FBA and NODDI, as well as the deep learning models that are used as comparisons for these experiments. However, moving the related work to the introduction would also be an appropriate place, and we are happy to do so.

---

> > > > > ### Author Response · Authors · 2023-08-11
> > > > >
> > > > > ### M.2 "Please add PSNR and SSIM..."
> > > > >
> > > > > Table 1 within the rebuttal PDF gives the PSNR values for the single-shell experiment (PSNR equivalent of Table 2 within the main manuscript). Table 1 shows a similar trend to the results demonstrated within the main manuscript, with the only difference of best-in-class going to the RCNN in $q_{\mathrm{in}} = 6, b = 1000$, instead of PCCNN-Bv. We will include PSNR and MSSIM tables for all experiments within the camera-ready manuscript.
> > > > >
> > > > > ## Limitations
> > > > >
> > > > > ### L.1 "The paper does not list any..."
> > > > >
> > > > > We agree with the reviewer's assessment of the limitations of the paper and therefore propose the following revision:
> > > > >
> > > > > Super-resolution within a medical imaging context presents inherent risks including hallucinations or incorrect predictions, particularly when making predictions on out-of-distribution data. Given that this work only includes data from relatively young healthy adults, future work will be needed to validate these methods in a diverse set of diagnostic settings, such as datasets with pathologies, a wide age range, and a variety of acquisition parameters.
> > > > >
> > > > > Modern deep learning frameworks rely on highly optimised subroutines to perform standard discrete convolutions. Subsequently, despite having a lower number of parameters, the PCCNNs inference and train times were longer than the other methods presented in this work. This is because the PCConv uses a bespoke implementation that does not benefit from the aforementioned subroutines. This highlights the need for future research into more efficient implementations of the PCConv operation.

---

> ### Comment · Area_Chair_e1ko · 2023-08-18
>
> Dear Reviewer bdM9,
>
> The author-reviewer discussion is closed on Aug 21st 1pm EDT, could you please read the rebuttal and give your final rating? Thanks so much!
>
> Best,
>
> AC

---

> ### Comment · Reviewer_bdM9 · 2023-08-18
> **Rebuttal response**
>
> Thank you for the extensive rebuttal and my apologies for the delay in response. It admirably addresses most points of concern raised in my review and I am raising my score from a 'borderline reject' to a 'borderline accept'.
>
> My score is not higher primarily due to limited evaluation concerns arising from weakness A1 i.e. only using 8 test subjects. While I agree that several comparable works also use limited sample sizes for evaluation, they typically have some signal amongst the results. There appears to be no clearly discernible pattern amongst the ablations in the primary results presented in the tables of the main text across the various modes of evaluation. I speculate that this is due to a too small sample set for effects to be clear. Cross-validation instead of a single static split might also reveal more interpretable results, but that would require large-scale reexperimentation that is infeasible for a rebuttal period.
>
> In fairness, for readers unfamiliar with its details, dMRI processing and preprocessing is severely time and compute intensive to a much greater extent than traditional volumetric data. It is thus understandable that a single static split with 8 evaluation subjects was used in the current submission. Nevertheless, I am uncertain as to what the quantitative takeaway is from the current version of the results. I am open to further discussion.

---

> > ### Author Response · Authors · 2023-08-20
> >
> > Thank you for revising the score in light of the continued discussion.
> >
> > We agree that the current analyses do not present a definitive signal with regards to the relative benefits of the PCCNN modifications. However, the modifications primarily serve to demonstrate the *flexibility* of the parametric continuous framework in incorporating additional coordinate information. The focus of the paper is how the PCCNN performs compared to baselines within this task. Here, the PCCNN family of models clearly outperform other comparable models, and this point is further demonstrated by the inclusion of the q-space CGAN as can be seen in Figure 2 of the rebuttal PDF.

---

### Author Rebuttal · Authors · 2023-08-09

We would like to thank the reviewers for taking the time to review our work in great detail. We look forward to further discussions.

The notable additions we have made during this rebuttal period include:

- A figure that visualises FODs reconstructed from different angular super-resolution models. This helps to qualitatively quantify the effects of applying the methods to this particular analysis.
- A figure with a new comparable dMRI angular super-resolution model (Q-space CGAN). In this figure, we demonstrate the difference in mean absolute error in an axial slice of a subject. In particular, we show that our method distinctly outperforms the Q-space CGAN.
- A table that demonstrates two additional ablation studies. One pertaining to the removal of b-vector information entirely from the network, as well as one looking at the effect of varying the hyperparameter $d_{\mathrm{max}}$.
- A table displaying the PSNR values for one of the experiments.
- Given the limited rebuttal period, we were unable to conduct extensive analyses. However, we have detailed further analyses that we will pursue for the camera-ready manuscript.

The additional figures and tables can be found in the rebuttal PDF.

---

### Decision · Program_Chairs · 2023-09-21

**Decision:**

Accept (poster)

**Comment:**

The paper received 4 (borderline) accept and 1 borderline reject. Reviewer PJ9o thinks that the approach combines existing ideas to seize this opportunity by creating a framework that convolves over both dense grid data and sparsely sampled spherical data concurrently, augmented by context/domain information. The authors provide additional experiments and analysis in the rebuttal, which addresses most concerns of the reviewers. Several reviewers (i.e., Reviewer Vyq4 and Kmgz) are convinced by the rebuttal and adjust their scores during the rebuttal phase.

Given the reviewers' comments and discussions with the authors, AC believes the paper can be accepted. It is recommended authors carefully consider reviewer comments and address them in the revision.